

# Scale and spatial distribution assessment of rainfall-induced landslides along mountain roads

Chih-Ming Tseng[1], Yie-Ruey Chen[1], Szu-Mi Wu[2]

[1]Department of Land Management and Development, Chang Jung Christian University, Tainan, 71101, Taiwan.
[2]Chen-Du Construction Limited, Taoyuan, 33059, Taiwan.

*Correspondence to*: Chih-Ming Tseng (cmtseng@mail.cjcu.edu.tw)

**Abstract.** This study focused on landslides along mountain roads that were caused by Typhoons Nanmadol (2011) and Kong-rey (2013). Image interpretation techniques were employed to interpret satellite images captured before and after the

typhoons to derive the changes in slope surfaces. The multivariate hazard evaluation method was adopted to establish a landslide susceptibility assessment model. We then mapped landslide susceptibility and locations to determine the relationship between the spatial distribution of landslide areas and the natural environment along mountain roads. The results can serve as a reference for preventing and mitigating slope disasters on mountain roads.

## 1 Introduction

Taiwan is an island of which three quarters of its land area consists of slope land that is 100 m above sea level, or is less than that but has an average gradient of 5% above (Soil and Water Conservation Bureau, 2012). Much of this sloped land has a steep gradient and fragile geological formations. Taiwan is hit by an average of 3.4 typhoons every year during years 1911 to 2016 (Central Weather Bureau, 2017). In additions, an average annual rainfall reach 2,506 mm in years 1949 to 2012 (Water Resources Agency, 2017). Typhoons usually occur between July and October, and 70%–90% of the annual rainfall is

composed of heavy rain directly related to typhoons (SWCB, 2012). Concentrated rainfall causes heavy landslides and debris flows every year (Dadson et al., 2004). The threat of disaster currently influences industrial and economic development and the road networks in endangers areas; thus, establishing disaster evaluation mechanisms is imperative.

Slope collapse analyses first investigate causal factors, or specifically, the combination of factors that cause landslides. These factors are then used to create landslide susceptibility models and estimate landslide potential. Landslide susceptibility

can be defined as the probability of the occurrence of a landslide in an area according to the relationship between the occurrence distribution and a set of predisposing factors such as geo-environmental thematic variables in the area (Guzzetti et al., 2005). In recent years, many studies have investigated the factors that cause sediment disasters (Mora and Vahrson, 1993; Fernandez, et al., 1999; Popescu, 2002; Wang and Sassa, 2006; Lee et al., 2008; Chen et al., 2009; Abay and Barbieri, 2012; Chue et al., 2015). However, the adequate number of causal factors required to assess slope conditions is unclear.



Some natural disasters are closely related to the characteristics of the natural environment; similarly, each landslide area is created by unique factors, which generally comprise several impact factors such as potential causes (e.g., geology, topography, and hydrology) and impetuses (e.g., rainfall, earthquakes, and anthropogenic factors) (Chen et al., 2013a; Chen et al., 2013b; Chue et al., 2015). Geological factors include lithological factors, structural conditions, and soil thickness;

topographical factors include slope, aspect, and elevation; and anthropogenic factors include deforestation, road construction, land development, mining, and alterations of surface vegetation (Chen et al., 2013a; Chen et al., 2013b; Chue et al., 2015). Studies have used various methods to evaluate or estimate the factors that influence landslide potential and have attempted to identify the more influential factors(Carrara et al., 1999, Guzzetti et al., 2006, Li et al., 2012, Bordoni et al., 2015). However, considerable uncertainty exists among causal factors, and this prevents problems that require solving from being

comprehensively understood. Nevertheless, predictions of landslide susceptibility and scale provide vital information. Early research analyzed the correlation between land use and altitude, slope gradient, and slope aspect in different natural environments (Miller and Sias, 1998). Although changes in land use accumulate over long periods, rapid environmental changes can still occur because of land development and localized natural damage.

Many studies have evaluated slope conditions based on an understanding of regional slope stability; researchers have

considered the influence of landslides on the surrounding environment and have established methods of evaluating regional landslide susceptibility assessment models (Yoshimatsu and Abe, 2006; Romeo et al., 2006; Van Westen et al., 2006). The methods used to analyze slope stability in previous studies can be broadly divided into two categories: topographical methods and engineering methods (Hansen, 1984). Topographical methods are quick, economical, and suitable for evaluating slope stability over large areas, and they are thus often applied in regional landslide susceptibility analysis

(Hansen, 1984). In this type of landslide susceptibility analysis, causal factors are collected and analyzed qualitatively or quantitatively. Regarding landslide evaluation, Yoshimatsu and Abe (2006) used the analytic hierarchy process (AHP) to assess landslide susceptibility in hazardous areas in Japan. In addition to using aerial photos to identify microlandform factors and areas with landslide potential, the researchers employed the AHP to issue scores and then used the total scores to determine landslide sensitivity. Romeo et al. (2006) utilized a quantitative approach to process landslide hazard and risk

assessments for large regions. The researchers analyzed landslide events, disaster proneness, and protected targets exposed to the threat of disaster, and they calculated the probabilities of landslide occurrence, spatial impact, and temporal impact. Van Westen et al. (2006) described the issues that could arise from using quantitative methods such as the date of landslide occurrence, types and number of landslides, and spatial and temporal probabilities to evaluate landslide risk over large areas. Several studies have used statistical methods to analyze landslide susceptibility. For example, Baeza and Corominas (2001)

examined susceptibility to shallow landslides in the mountainous areas of Spain by using discriminant analysis. Dai and Lee (2002) extracted seven factors including lithology, gradient, aspect, and elevation to conduct logistic regression, calculate susceptibility values, and map the landslide susceptibility of Lantau Island in Hong Kong. Ohlmacher and Davis (2003) utilized two factors, namely lithology and slope, as independent variables for logistic regression to assess landslide occurrence probability in Kansas, United States. Researchers have also applied the multivariate hazard evaluation method



(MHEM) (Su et al., 1998) to map landslide susceptibility in catchment areas (Lin et al., 2009). The MHEM is a nonlinear mathematical model that presents an instability index to indicate risk in different areas (Lin et al., 2009).

In some studies, landslide susceptibility analyses have focused on man-made facilities such as roads and railroads and have examined the landslide susceptibility of surrounding environments. For example, Das et al. (2010) used logistic regression to

create a landslide susceptibility assessment model for landslide-prone national highway road sections in the northern Himalayas in India, and they employed geotechnical-based slope stability probability classification methodology for verification. Their results indicated that logistic regression slightly underestimated the high landslide susceptibility. In a subsequent study, Das et al. (2012) used Bayesian logistic regression (BLR) to establish a landslide susceptibility assessment model for the road corridors in the Bhagirathi River valley of the Indian Himalayas, and they compared the results with those

of the logistic regression in their previous study. The BLR approach exhibited higher prediction performance. Martinović et al. (2016) used logistic regression to develop a model to evaluate the susceptibility of engineered slopes to shallow slides in the Irish Rail network. The aforementioned studies on the landslide susceptibility of areas surrounding man-made facilities have not investigated characteristics such as the location and scale of landslides occurring in upper or lower slopes, and such characteristics thus constituted one of the objectives of the present study.

Technological progress has provided various advanced tools and techniques for land use monitoring. In recent years, aerial photos or satellite images have been commonly used in post disaster interpretations and assessments of landslide damage on large-area slopes (Erbek et al., 2004; Lillesand et al., 2004; Nikolakopoulos et al., 2005; Lin et al., 2005; Chen et al., 2009; Otukei and Blaschke, 2010; Chen et al., 2013a). Satellite images offer the advantages of short data acquisition cycles, swift understanding of surface changes, large data ranges, and being low cost, particularly for mountainous and inaccessible areas.

With the assistance of computer analysis and geographic information system (GIS) platforms, researchers can quickly determine land cover conditions. Thus, satellite images are suitable for investigating large areas and monitoring temporal changes in land use (Liu et al., 2001). Satellites can capture images of the same area multiple times within a short period; the images are consistent in quality and are digitized, rendering them convenient for computer applications. Studies have indicated that land surface change detection is the process of exploring the differences between images captured at different

25   times. With multispectral satellite images, land surface interpretations involve comparisons of multitemporal images that are completely geometrically aligned (Liu et al., 2001; Chadwick et al., 2005; Chen et al., 2009; Chue et al., 2015).

Based on studies regarding historical disasters (National Science and Technology Center for Disaster Reduction [NCDR], 2017), we selected part of Provincial Highway No. 20 in the catchment area of Laonong River in southern Taiwan as our study area. Regarding time, we focused on periods before and after landslides that occurred in the study area as a result of

Typhoon Nanmadol (2011) and Typhoon Kong-rey (2013). We applied the maximum likelihood method to interpret and categorize high-resolution satellite images, thereby determining the land surface changes and slope disasters in the study area before and after the rainfall events. By using a GIS platform, we constructed a database of the rainfall and natural environment factors. Subsequently, we developed a landslide susceptibility assessment model by using the MHEM and extracted the locations of landslide areas to explore the relationship between the natural environment and the spatial





distribution of these areas. The results of this study could serve as a reference for the prevention and mitigation of slope disasters on hillsides along mountain roads.

## 2 Methodology

### 2.1 Maximum likelihood

The maximum likelihood classifier is a supervised classification method (SCM). SCMs include three processing stages: training data sampling, classification, and output. The underlying principle of supervised classification is the use of spectral pattern recognition and actual ground surface data to determine the types of data required and subsequently select a training site, which has a unique set of spectral patterns. To accurately estimate the various spectral conditions, the spectral patterns of the same type of feature are combined into a coincident spectral plot before the class of the training site is selected. Once

training has been completed, the entire image is classified based on the spectral distribution characteristics of the training site by using statistical theory for automatic interpretation (Lillesand et al., 2004).

To facilitate the calculation of probability in the classification of unknown pixels, the maximum likelihood method assumes a normal distribution in the various classes of data. Under this assumption, the data distribution can be expressed using covariance matrices and mean vectors, both of which are used to calculate the probability of a pixel being assigned to a land

cover class. In other words, the probability of $X$ appearing in class $i$ is calculated using Eq. (1), and the highest probability is used to determine the feature of each pixel (Lillesand et al., 2004).

$$p(X|C_i) = (2\pi)^{d/2} |\Sigma_i|^{-1/2} \exp\left[-\frac{1}{2}(X - \mu_i)^T \Sigma_i^{-1}(X - \mu_i)\right] \tag{1}$$

$$x = \begin{bmatrix} x_1 \\ x_2 \\ \vdots \\ x_d \end{bmatrix} \qquad \mu_i = \begin{bmatrix} \mu_{1i} \\ \mu_{2i} \\ \vdots \\ \mu_{di} \end{bmatrix} \qquad \Sigma_i = \begin{bmatrix} S_{11} & S_{12} & \cdots & S_{1d} \\ S_{21} & S_{22} & \cdots & S_{2d} \\ \vdots & \vdots & \ddots & \vdots \\ S_{d1} & S_{d2} & \cdots & S_{dd} \end{bmatrix}$$

In this equation,

$d$ denotes the number of features;

$X$ denotes a sample expressed using features and has $d$ dimensions;

$p(X|C_i)$ denotes the probability that $X$ originates from class $i$;

$\Sigma_i$ denotes the covariance matrix of class $i$;

$\Sigma_i^{-1}$ denotes the inverse matrix of $\Sigma_i$;

$|\Sigma_i|$ denotes the determinant of $\Sigma_i$;

$\mu_i$ denotes the mean vector of classification $i$;

$(x - \mu_i)^T$ denotes the transpose matrix of $(x - \mu_i)$; and



$S_{ij}$ denotes the covariance of classes $i$ and $j$.

During classification, the maximum value of the probability density functions of sample $X$ in each class is used to determine which class the sample belongs to. The maximum likelihood classification decision is shown in Eq. (2).

$$X \in C_m, \qquad m \subset \{1,2,\cdots,k\}$$

*if*

$$p(X|C_m) = \max\{p(X|C_i), \quad j = 1,2,\cdots k\} \tag{2}$$

The question in classification is how to effectively separate the classes in the feature space, or in other words, how to divide the feature space. Maximum likelihood is a common approach that offers fairly good classification accuracy (Bruzzone and

10 Prieto, 2001; Chen et al., 2004). Thus, we adopted maximum likelihood to interpret and classify the satellite images.

## 2.2 MHEM

The MHEM is a diverse non-linear mathematical model. Based on relative relationships, the MHEM presents an instability index ($D_t$) to indicate risk in different areas. The objective is to analyze the variability of landslide-inducing factors and estimate the variance of causal factors to derive a suitable landslide susceptibility assessment model (Lin et al., 2009; Chue,

et al., 2015).

The causal factors in the MHEM are rated based on the frequency of landslide occurrence, which is calculated as follows:

$$R_i = \frac{r_i}{r_T} \tag{3}$$

where $R_i$ represents the landslide grid ratio of the various factors in grade $i$, $r_i$ represents the number of landslide grids in grade $i$, and $r_T$ represents the total number of grids. Thus, landslide percentage $X_i$ is expressed as

$$X_i = \frac{R_i}{\sum R_i} \tag{4}$$

where $X_i$ denotes the landslide percentage of grade $i$ and $\Sigma R_i$ denotes the sum of the landslide grid ratios.

To estimate the weight of influence of each causal factor, the coefficient of variation ($V$) of the landslide ratios derived from the grade of the impact factors is used to represent the sensitivity of landslide ratios in different impact factor grades. A smaller coefficient of variation denotes higher similarity among the landslide probabilities in the various grades, which

indicates that this factor grading method cannot determine which areas have higher or lower landslide probabilities. By contrast, a greater coefficient of variation denotes that this factor grading method can be used to describe the influence of factor grades on landslides. Thus, the coefficient of variation among the impact factors can indicate the factor weights. The coefficient of variation is calculated as shown in Eq. (5):





$$V = \frac{\sigma}{X} \times 100\% \qquad (5)$$

where $\sigma$ is the standard deviation and $X$ is the mean landslide percentage of the various factor grades.

We divided the coefficient of variation of each individual factor by the total coefficient of variation of all factors to derive the factor weight, which represented the degree of influence of the factor on landslide occurrence. The factor weight can be
calculated as shown in Eq. (6), where $W$ is the factor weight and $V$ is a coefficient of variation.

$$W_i = \frac{V_i}{V_1 + V_2 + \cdots + V_n} \qquad (6)$$

The graded score value of the causal factors can be calculated using Eq. (7), based on the landslide percentages of the various factor grades, and presented in relative values ranging from 1 to 10.

$$d_n = \frac{9(X_i - X_{min})}{(X_{max} - X_{min})} + 1 \qquad (7)$$

In Eq. (7), $X_i$ represents the causal rate of the sample region and $X_{max}$ and $X_{min}$ represent the maximum and minimum landslide percentages of the factor in the various sample regions, respectively.

Finally, the weight $(W_i)$ of each factor is determined by the rank of its variance $(V)$, and each factor is assigned a different weight. Subsequently, a nonlinear mathematical model can be derived as follows:

$$D_t = d_1^{W_1} \times d_2^{W_2} \times d_3^{W_3} \times d_4^{W_4} \times d_5^{W_5} \cdots \cdots \times d_n^{W_n} \qquad (8)$$

where $D_t$ is the instability index of the samples, expressed using relative values ranging from 1 to 10. A cumulative value closer to 10 indicates greater landslide potential, whereas a cumulative value closer to 1 indicates lower landslide potential.

### 2.3 Accuracy assessment

This study employed the aforementioned maximum likelihood method to classify satellite images. To determine whether the accuracy of image classification was acceptable, we adopted an error matrix to test for accuracy. An error matrix is a square
matrix that presents error conditions in the relationship between ground surface classification results and reference data (Verbyla, 1995). Such a matrix contains an equal number of columns and rows, and the number is determined by the number of classes. For example, Table 1 contains four classes. The columns show the reference data, and the rows show the classification results. The various elements in the table indicate the quantity of data corresponding to each combination of classes.

In the table, $X_{12}$ represents the amount of data that was interpreted as Class A but actually belongs to Class B, whereas $X_{21}$ indicates the amount of data that was interpreted as Class B but actually belongs to Class A. $X_{11}$ and $X_{22}$ represent the amount of data accurately classified as Class A and Class B, respectively. An error matrix is generally used to check the quality of





classification results in statistics (Congalton, 1991; Verbyla, 1995). In the present study, we evaluated the accuracy of the classification results based on the overall accuracy (*OA*) and *Kappa* value (Cohen, 1960), which is the coefficient of agreement derived from the relationship between the classification results and training data. These two parameters are explained as follows.

### 2.3.1 *OA*

*OA* is the simplest method of overall description. For all classes, *OA* represents the probability that any given point in the area will be classified correctly.

$$OA = \left[ \frac{1}{N} \Sigma_{i=1}^{n} X_{ii} \right] \times 100\% \tag{9}$$

In Eq. (9), *N* denotes the total number of check points and $X_{ii}$ denotes the number of correctly classified checkpoints.

### 2.3.2 *Kappa* coefficient

The *Kappa* ($\hat{K}$) coefficient indicates the degree of agreement between the classification results and reference values and shows the percentage reduction in the errors of a classification process compared with the errors of a completely random classification process. Generally, the *Kappa* coefficient ranges from 0 to 1, and a greater value indicates a higher degree of agreement between the two sets of results, as shown in Eq. (10):

$$\hat{K} = \frac{N \Sigma_{i=1}^{n} X_{ii} - \Sigma_{i=1}^{n} \left( X_{i+} \times X_{+i} \right)}{N^2 - \Sigma_{i=1}^{n} \left( X_{i+} \times X_{+i} \right)} \times 100\% \tag{10}$$

As reported by Landis and Koch (1977), a *Kappa* coefficient greater than 0.8 signifies a high degree of accuracy, whereas a coefficient between 0.4 and 0.8 or less than 0.4 indicates moderate or poor accuracy, respectively.

### 2.4 Rainfall analysis method

In previous studies regarding the influence of rainfall on landslides, rainfall intensity and accumulated rainfall have been most commonly used as causal factors of landslides (Giannecchini, 2006; Chang et al., 2007; Giannecchini et al., 2012; Ali et al., 2014). Therefore, we adopted effective accumulated rainfall (*EAR*) and intensity of rolling rainfall ($I_R$) as rainfall indices and impact factors of landslides in the present study. These two indices are explained as follows.

### 2.4.1 *EAR*

Generally, rainfall is considered the trigger of slope collapse, whereas previous rainfall can be regarded as a potential factor of a landslide. Previous rainfall influences the water content of the soil, which in turn affects the amount of rainfall required to trigger a landslide (Seo and Funasaki, 1973).





Figure 1 shows an illustration of rainfall events defined based on *EAR* (Seo and Funasaki, 1973). The diagram shows a concentrated rainfall event with no rainfall in the preceding or subsequent 24 hours and can thus be considered a continuous rainfall event. A continuous rainfall event that occurs simultaneously with a landslide is the main rainfall event. The beginning of the main rainfall event is defined as the time point when the rainfall first reaches 4 mm. The calculation of
accumulated rainfall ends at the time when the landslide occurs. However, because the exact time of a landslide cannot be precisely determined, we regarded the hour with the maximum rainfall during the main rainfall event as the time when the landslide occurred in this study.

In accordance with previous studies, we defined *EAR* as the sum of direct and previous indirect rainfall. Previous indirect rainfall is the rainfall accumulated during the 7 days prior to the main rainfall event and can be expressed as follows (Seo
and Funasaki, 1973):

$$\sum_{n=1}^{7} k^n P_n = P_b \tag{11}$$

where $P_b$ denotes the previous indirect rainfall, $P_n$ denotes the rainfall during the *n* days prior to the main rainfall event (mm), and *k* denotes a diminishing coefficient set as 0.9 in this study. Direct rainfall encompasses the continuous rainfall accumulated during the rainfall events, starting from the first rainfall to the time of landslide occurrence. Direct rainfall has a
direct and effective impact on landslide occurrence and is thus not diminished. Therefore, *EAR* could be expressed as follows in this study:

$$EAR = P_r + P_b \tag{12}$$

where $P_r$ represents the rainfall accumulated during the main rainfall event from the first rainfall to the time of landslide occurrence, and $P_b$ represents the previous indirect rainfall.

**2.4.2 $I_R$**

Rainfall intensity refers to the amount of rainfall within a unit of time. It is considered a crucial index for evaluating disasters because greater intensity or longer durations have considerable impacts on slope stability. Furthermore, rainfall-induced landslides may be triggered by several hours of continuous rainfall. The raw rainfall data in this study was hourly precipitation; thus, $I_R$ can be expressed as follows:

$$I_{mR} = \sum_{t-m+1}^{m} I = I_{t-m+1} + I_{t-m+2} + \cdots + I_t \tag{13}$$

where *I* denotes rainfall intensity, *m* denotes the number of rolling hours of rainfall (set as 3 hours in this study), $I_{mR}$ denotes the $I_R$ during *m* hours, and $I_t$ denotes the rainfall intensity during hour *t*.





## 3 Study area

We referred to the historical data on road disasters from the NCDR (2017) and considered road sections where rainfall-induced collapses occurred frequently in southern Taiwan. We focused on the periods before and after Typhoon Nanmadol (2011) and Typhoon Kong-rey (2013) hit southern Taiwan, and we selected part of Provincial Highway No. 20 in the

catchment area of Laonong River in southern Taiwan as our study area (Fig. 2), which includes areas from three districts in Kaohsiung City (Jiasian, Liouguei, and Tauyuan).

## 4 Image interpretation and classification

### 4.1 Preprocessing of satellite images

This study employed and interpreted satellite images taken by FORMOSAT-2 (FM2). FM2 images have been extensively

used to identify natural disasters and land use (e.g., Lin et al., 2004; Lin et al., 2006; Liu et al., 2007; Chen et al., 2009, Lin et al., 2011; Chen et al., 2013a). In the present study, prior to interpretation, the satellite images underwent spectral fusion, coordinate positioning, cropping, and cloud removal. The images taken by FM2 are multispectral with blue, green, red, and near-infrared (NIR) wavelengths (Chen et al., 2013a; Chue et al., 2015). Image fusion and coordinate positioning were conducted using the import data and coordinate positioning tool of ERDAS IMAGINE (2013). Because clouds and shadows

affect the accuracy of image interpretations, we used the image analysis tool of ArcGIS to remove clouds from the images.

### 4.2 Training site selection and mapping

To map the sample areas required for image interpretation, we overlapped the high-resolution, preprocessed satellite images of the study area before and after the typhoons and mapped the training sites by using a GIS platform. Based on field investigations and relevant studies (Chen et al., 2009; Chen et al., 2013a; Chue et al., 2015), we selected areas with water,

roads, buildings, crops, vegetation, river channels, and bare land within the study area as the sample area factors for interpretation training.

### 4.3 Image interpretation and accuracy assessment

Image interpretation and classification were conducted using the Maximum Likelihood module in ERDAS IMAGINE. The interpretation and classification results of the satellite images before and after Typhoon Nanmadol in 2011 and Typhoon

Kong-rey in 2013 are shown in Fig. 3. The different colors in the images represent different interpretation factors.

To verify the accuracy of the results, we randomly extracted 25 points from the satellite images for each training factor as checkpoints and tested the accuracy by using the aforementioned error matrix approach. With the satellite images before and after Typhoon Kong-rey in 2013 as an example, Table 2 shows the error matrix and accuracy assessment results of the satellite image interpretation and classification processes. Table 3 presents the *Kappa* values and *OA* results of the satellite



images captured before and after the two typhoons. As mentioned, *Kappa* values ranging from 0.4 to 0.8 indicate moderate accuracy, and thus the interpretation results had moderate to high accuracy.

## 5 Landslide susceptibility assessment of slopes along mountain roads

To evaluate the landslide susceptibility of slopes within the study area, we constructed 8 m × 8 m grids by using the GIS

platform along with the interpretation results of the two typhoons. We also constructed an 8 m × 8 m digital elevation model (DEM) and input the classification results, map of the natural environment, and rainfall data into the grid to aid subsequent landslide susceptibility assessments.

### 5.1 Impact factor selection and factor correlation test

### 5.1.1 Impact factor selection

Referring to Chen et al. (2009), we divided the impact factors of landslides into three categories: natural environment, land disturbance, and rainfall.

A. Natural environment factors

(A) Elevation

The influence of elevation varies with the climate and thus affects the distribution of vegetation on the slope and type of

weathering. In addition, elevation reflects the influence of geological structure, stress, and time. The highest and lowest elevations in the study area were 1480.6 and 365.2 m, respectively. Using the GIS platform, we extracted the elevation data from the DEM of the study area to estimate the mean elevation of each grid. We divided the elevation data into seven grades at intervals of 300 m.

(B) Slope gradient

A slope's gradient generally exerts significant impact on slope stability. By using the DEM and gradient analysis of the GIS platform, we calculated the mean gradient of each grid in the study area; subsequently, we divided the gradient values in the grids within the study area into seven grades.

(C) Aspect

Rainfall-induced landslides are subject to the influence of seasonal changes such as those related to rainfall and wind

direction. Thus, the direction of the slope must be considered. As described, we used the DEM and aspect analysis function of the GIS platform to calculate the average aspect of the grids in the study area. According to their direction, we divided them into six categories from windward to flat ground.

(D) Geology




Referring to the digital file of the Geologic Map of Taiwan, Scale 1:50,000, Chiahsien, which was compiled by the Central Geological Survey of the Ministry of Economic Affairs in 2000, we determined that the geology of the study area includes five types of rock: the upper part of Changshan Formation, the Tangenshan Formation, the Changchihkeng Formation from the Miocene period, and modern alluvium and terrace deposits from the Holocene period. We divided geological strength into six grades (Chen et al., 2009).

(E) Terrain roughness

Terrain roughness refers to the degree of change in grid height. Wilson and Gallant (2000) proposed the use of the standard deviation of height within a radius to measure the degree of change in height because of its indicative meaning in relation to changes in regional height. Using the analysis function in ArcGIS, we calculated the terrain roughness of the DEM. Statistical cluster analysis was used to automatically divide terrain roughness into six grades.

(F) Slope roughness

Slope roughness refers to the fluctuations in slope gradient in the grids. High slope roughness means that the slope gradient varies considerably (Wilson and Gallant, 2000). Slope roughness is calculated through the same method as terrain roughness, except with the original elevation values being replaced with the slope gradient values obtained using ArcGIS. Just as terrain roughness was graded, we first used the analysis function of ArcGIS to estimate the slope roughness of each grid, after which we used cluster analysis to automatically divide slope roughness into six grades.

(G) Distance to water

We calculated the distances to water using ArcGIS and divided the distances into seven levels.

(H) Distance to road

We calculated the distances to road using ArcGIS and divided the distances into seven levels.

B. Land disturbance factors

Land disturbance varies with space and time. We made some revisions to the approach proposed by Chen et al. (2009, 2013b) to calculate land disturbance and selected roads, buildings, crops, bare land, and vegetation as the land disturbance factors of landslides in the study area. We extracted the disaster and ground surface data from previous satellite image interpretation and classification results and input the land disturbance factors into the grids by using the GIS platform. The scores of the index for disturbance condition ($I_{DC}$) in the grids are shown in Table 4.

C. Rainfall factors

We collected precipitation data from weather stations close to the Central Weather Bureau, including Guanshan, Biaohu, Hsiao Guanshan, Gaojhong, Sinfa, Jiasian, and Xi'nan. Table 5 displays the station information. We then calculated the *EAR* and 3-hour $I_R$ ($I_{3R}$) levels observed at each station. The results from Typhoon Nanmadol in 2011 and Typhoon Kong-rey in 2013 are compiled in Table 6. By using the Inverse Distance Weighting (*IDW*) function of ArcGIS and the *EAR* and





maximum $I_{3R}$ values of the weather stations, we estimated the rainfall of each grid throughout the study area and then used cluster analysis to divide the results into six levels.

### 5.1.2 Factor correlation test

To establish a landslide susceptibility assessment model, we selected elevation, slope gradient, aspect, geology, terrain roughness, slope roughness, distance to water, distance to road, $I_{DC}$, and rainfall as landslide-triggering factors. Rainfall included *EAR* and maximum $I_{3R}$.

We employed the Pearson correlation test tool in SPSS (2005) to examine the correlation among these factors. The correlation coefficients ranged from -1 to +1, with +1, -1, and 0 indicating complete positive correlation, complete negative correlation, and no correlation between two variables, respectively. Factors with high correlation were then subjected to a paired sample *t* test conducted using SPSS to examine the significance of the correlation between them. Those with high correlation were eliminated.

Table 7 presents the test results regarding the correlation between the impact factors. As shown, the degree of correlation between most factors was moderate to low. A high degree of correlation was found only between elevation and terrain roughness and between slope gradient and slope roughness. Thus, we administered paired sample *t* tests to these two factor pairs to test the significance of the correlation. The results in Table 8 show that the significance was 0 (<0.05) for the correlation between both pairs, indicating no correlation; thus, these factors were not eliminated.

### 5.2 Landslide susceptibility assessment and hazard map

To apply the MHEM to establish a landslide susceptibility assessment model, we input the natural environment, land disturbance, and rainfall factors into the grids by using the GIS platform. By using the changes in bare land between the images before and after the typhoons and applying image subtraction aided by manual checking, we obtained the grid data of the rainfall-induced landslide locations in the study area. With the study area after Typhoon Nanmadol in 2011 as an example, we considered *EAR* during the rainfall period and rated the grades by using the factor weights derived using the MHEM, as shown in Table 9.

The calculation process is explained in this paper by using elevation as an example. In accordance with factor selection, the elevation factor was divided into seven grades. Aided by the GIS platform, we calculated the total number of grids, total number of landslides, and landslide percentage within each elevation level in the study area by using Eqs. (3) and (4). We then calculated the standard deviation, coefficient of variation, and weight values by using Eqs. (5) and (6); the results are listed in Table 9. The presented results show that the standard deviation ($\sigma$), coefficient of variation ($V$), and factor weight ($W$) of landslide percentage were 0.021, 0.764, and 0.087, respectively. Based on the landslide percentages of the elevation factor and the minimum and maximum landslide percentages, we subsequently obtained the scores of the factors by using Eq. (7). Finally, we calculated the instability indices by using the weight values and scores of the factors through Eq. (8). Furthermore, the results in Table 9 indicate that the degrees of land disturbance ($I_{DC}$), geology ($G_s$), slope gradient ($S_s$), and





slope roughness had the greatest influence on landslides in the study area, followed by distance to water ($D_s$), *EAR*, and elevation ($E_l$).

We considered *EAR* and $I_{3R}$ and used an instability index to determine the level of landslide susceptibility of slopes throughout the study area. The derived instability index intervals (Table 10) ranged from 2.05 (2.02) to 9.59 (9.96). By using the concept of log-normal distribution in statistics, we converted the levels of landslide susceptibility derived using the MHEM into probabilities of landslide occurrence. The calculation formula of the log-normal distribution is shown in Eq. (14):

$$P(F) = \frac{1}{x\sigma\sqrt{2\pi}} e^{-\frac{1}{2}[(\ln x - \mu)/\sigma]^2} \tag{14}$$

where $x$ denotes the level of the instability index and $\mu$ and $\sigma$ denote the mean and standard deviation of the level of the instability index, respectively. After calculating the probabilities of landslide occurrence by using the log-normal distribution, we normalized the probabilities to range from 0 to 1 for convenience. The normalization formula is shown in Eq. (15). The landslide probability intervals calculated based on *EAR* and $I_{3R}$ are presented in Table 10.

$$P(F)' = \frac{(X_i - X_{min})}{(X_{max} - X_{min})} \tag{15}$$

In Eq. (15), $X_i$ represents the factor being normalized and $X_{max}$ and $X_{min}$ represent the maximum and minimum values of the factor, respectively.

We employed the mean probability of landslide occurrence to differentiate between high and low landslide susceptibility. Landslides were considered more likely to occur in areas where the probability of landslide occurrence was greater than the mean. By contrast, landslides were considered less likely to occur in areas where the probability of landslide occurrence was lower than the mean. With rainfall factor *EAR* as an example, we determined the mean probability of landslide occurrence to be 0.46. We further divided landslide susceptibility into four levels: high (0.731–1), medium high (0.461–0.73), medium low (0.23–0.46), and low (0–0.23). The results showed that the mean probability of landslide occurrence varied little, regardless of whether it was calculated using *EAR* or $I_{3R}$.

By using the GIS platform, we considered the landslide susceptibility calculated using *EAR* for Typhoon Nanmadol in 2011 as an example. As illustrated in Fig. 4, we included an overlay created by the NCDR and showing the locations of historical disasters within the study area. The results revealed a total of 24 historical disasters, 17 of which were situated in areas of medium high or high landslide susceptibility. Therefore, the estimation accuracy in this study was approximately 71%. Regarding Typhoon Kong-rey in 2013, 18 historical disasters occurred within areas of medium high or high landslide susceptibility, thereby yielding 75% accuracy. Table 11 presents the accuracy levels associated with using different rainfall factors to calculate landslide susceptibility for different typhoons.





# 6 Landslide location analysis

We analyzed the spatial characteristics of landslides by using landslide locations collected from before and after both of the two typhoons and the land surface interpretation results of the study area.

## 6.1 Investigation of landslide impact factors and landslide area

The influence of causal factors on landslides varies. In this study, we examined the relationships between landslide area and various landslide factors. By using the area of landslides (i.e., the number of landslide grids) induced by Typhoon Nanmadol in 2011 as an example, we investigated the influences of the causal factors (elevation, slope gradient, aspect, geology, slope roughness, terrain roughness, distance to water, distance to road, and degree of land disturbance) on landslides. The various factor levels and corresponding numbers of landslide grids are shown in Fig. 5a–i.

Figure 5a presents the relationship between different levels of elevation and the number of landslide grids (landslide area). As shown in the figure, the number of landslide grids in the study area peaked at elevations between 450 and 750 m and then declined as the elevation increased. Figure 5b displays the relationship between different levels of slope gradient and the number of landslide grids (landslide area). As shown in the figure, the number landslide grids in the study area increased with the slope gradient and peaked between 30° and 55°. Landslides rarely occurred on slopes steeper than 55°. Figure 5c

illustrates the relationship between aspect and the number of landslide grids, with aspect divided into eight categories: north, northeast, east, southeast, south, southwest, west, and northwest. As shown in the figure, the number of landslide grids was highest on slopes facing south, followed by those on slopes facing east and southeast. We speculate that this is because rainfall during the typhoon season in Taiwan promotes poor cementation and high weathering on slopes along rivers, which consequently prompts these slopes to develop toward low-lying rivers (which run from the northeast to the southwest) after

rainfall events.

Figure 5d shows the relationship between geology and the number of landslide grids. As shown in the figure, the Sanhsia Group and its stratigraphic equivalence lead to landslides more easily than does the Lushan Formation in the study area. The Sanhsia Group and its stratigraphic equivalence mainly comprise sandstone, shale, and interbedded sandstone and shale. Shale has weaker cementation, lower strength, and a greater tendency to weather and fracture. By contrast, the Lushan

Formation consists of argillite, slate, and interbedded argillite and sandstone, and its strength is controlled by cleaving; some areas are prone to weathering and fracturing. Thus, both rock types are more likely to collapse, but on the whole, the Sanhsia Group and its stratigraphic equivalence collapse more easily than does the Lushan Formation. Furthermore, this result indicates that the locations of landslide areas within the study area are associated with geology. Figure 5e presents the relationship between slope roughness and the number of landslide grids. The number of landslide grids within a level of

slope roughness first increased with the slope roughness and then began to decline once a certain level of slope roughness (35–40) was reached. This result is similar to that of the influence of slope gradient on the number of landslide grids. Figure 5f displays the relationship between terrain roughness and the number of landslide grids. As shown in this figure, the results





are similar to those regarding the influence of elevation on the number of landslide grids; the number of grids declined when the terrain roughness was greater than 500 and was very small low the terrain roughness was greater than 1200.

Figure 5*g* illustrates the relationship between distance to water and the number of landslide grids. The presented results show a significantly greater number of landslide grids within 300 m of water. The width of the river channel within the study area was determined to range from 100 to 200 m, revealing that the development of landslide areas near water in the study area is caused by rainfall significantly raising the water level in the river, which scours the slope toe, affects slope stability, and triggers landslides. Figure 5*h* presents the relationship between distance to road and the number of landslide grids. The presented results reveal that areas between 100 and 300 m from roads had the greatest number of landslide grids. Further examination of the relationship between distance to road and the area and number of landslides revealed that most landslides between 0 and 100 m from roads were small collapses, whereas those between 100 and 300 m from roads were larger in area. The number of landslides 0–100 m from roads was greater than that 100–300 m from roads.

The degree of land disturbance can represent the degree of slope development. A greater degree of land disturbance likely indicates a greater degree of human development, which can yield a greater number of landslide grids. Figure 5*i* shows the relationship between the degree of land disturbance and the number of landslide grids. The presented results indicate that the number of landslide grids increased with the degree of land disturbance.

## 6.2 Investigation of rainfall factors and instability index

To understand the relationship between the rainfall factors and the degree of instability on the slopes in the study area after typhoons, we first removed the cloud cover grids from posttyphoon images and subsequently employed cluster analysis to divide the instability indices of the grids into three levels: high, medium, and low. We then collected random samples based on the proportions of landslide and nonslide grids in each level (50 landslide and 50 nonlandslide grid points) and plotted their relationship. Table 12 and Fig. 6*a–d* present the relationships between the rainfall factors ($EAR$ and $I_{3R}$), instability index, and landslide occurrence in the grids following Typhoon Nanmadol in 2011 and Typhoon Kong-rey in 2013. Figure 6*a* and *b* consider $EAR$, whereas Fig. 6*c* and *d* consider $I_{3R}$. The presented results indicate that the typhoon events increased the degree of slope instability ($D_t$) and landslide occurrence, regardless of whether $EAR$ or $I_{3R}$ was considered. Furthermore, significantly more landslide points were situated in areas of high instability than in areas of other levels of instability, and landslides rarely occurred in areas of low instability. Moreover, areas of high slope instability were prone to landslides even if their $EAR$ or $I_{3R}$ was low. By contrast, areas of low instability required more rainfall for landslides to be possible. The results (Table 12) further showed that the $EAR$ and $I_{3R}$ levels of Typhoon Kong-rey in 2013 were greater than those of Typhoon Nanmadol in 2011. Thus, in any $D_t$ level, the proportion of landslides that occurred in the study area after Typhoon Kong-rey was higher than that after Typhoon Nanmadol. Figure 6*e* and *f* present the relationships between $EAR \times I_{3R}$, the instability index, and landslide occurrence; $EAR \times I_{3R}$ is the index of rainfall-induced landslide (ILR), with a higher value indicating higher susceptibility to a landslide. The figures show that for a high instability index, even a small rainfall event





could trigger a landslide (lower right corners of the figures). By contrast, for a low instability index, a larger rainfall event could not easily trigger a landslide (upper left corners of the figures).

## 6.3 Landslide scale and spatial distribution

We employed the terrain tool in ERDAS IMAGINE and the DEM to identify the ridges and valleys in the study area.

Following the methods in previous studies (Meunier et al., 2008; Chue et al., 2015), we extracted the distances between the highest point of a landslide area and the nearest ridge ($dr$), between the lowest point of the landslide area and the nearest stream ($ds$), and between the ridge and the stream ($dt$) (Fig. 7). Furthermore, in Taiwan, many slopes are visible on developed, mountain roads built between ridges and streams. Therefore, we explored the spatial distribution of landslides above and below mountain roads. Similar to Fig. 7a, to explore the spatial distribution of landslides, we extracted the

distances between the highest point of a landslide area on a slope above a road and the nearest ridge ($dr$), between the lowest point of the landslide area and the nearest mountain road ($dmu$), and between the ridge and the mountain road ($dtu$) (Fig. 7b); we also investigated this distribution by extracting the distances between the highest point of a landslide area on a slope below a road and the nearest mountain road ($dmd$), between the lowest point of the landslide area and the nearest stream ($ds$), and between the mountain road and the stream ($dtd$) (Fig. 7c).

This study examined the spatial distribution of landslides in the region along Provincial Highway No. 20 before and after Typhoon Nanmadol in 2011 and Typhoon Kong-rey in 2013. Using the approach shown in Fig. 7a, we mapped the bare land in the study area, as shown in Fig. 8a–d. Of these figures, Fig. 8a and c show the conditions before the typhoons, whereas Fig. 8b and d present the conditions after the typhoons. The presence of bare locations near the $Y$ axis ($dr/dt \approx 0$) denotes that the bare land originated near the ridge. By contrast, the presence of bare locations near the $X$ axis ($ds/dt \approx 0$) denotes that the

bare land progressed toward the stream. Thus, the presence of bare locations near the origin denotes that the bare land originated near the ridge and progressed toward the stream.

The results in Fig. 8a–d show more bare locations in the lower right halves of the graphs, some of which are larger in area. The figures indicate fewer bare locations in the upper left halves of the graphs, and the ones that are present are smaller in area. These spatial distribution characteristics are similar to those derived by Meunier et al. (2008). We speculate that this is

25 because the frequency of rainfall-induced landslides increases significantly because of bank erosion, which is shown in the lower right half of Fig. 8 ($dr/dt \geq 0.5$ and $ds/dt \leq 0.5$). Furthermore, the bare locations before and after Typhoons Nanmadol and Kong-rey show that the bare land does not increase in number but increases significantly in area, implying that old landslides may result in more collapses or expansions of the affected area. In addition, the number of old landslides is greater than that of new landslides.

We explored the spatial distribution of landslides on slopes above (Fig. 9) and below (Fig. 10) mountain roads in the study area before and after Typhoon Kong-rey in 2013. Figure 9a and Fig. 10a present the spatial distribution of bare land before the typhoon, whereas Fig. 9b and Fig. 10b present the spatial distribution of bare land after the typhoon.





As shown in Fig. 9, most landslides on the slopes above the mountain roads occurred close to the roads, most likely because road construction involves cutting the slope toe and increasing the gradient. After the typhoon, the bare locations on the slopes above the roads in the study area did not increase in number significantly; thus, rainfall did not exert a substantial impact on the slopes above the roads. The results in Fig. 10 show bare locations on the slopes below the mountain roads

developing from near the roads to the streams. The bare locations near the streams may also have been affected by rainfall-induced bank erosion. However, the bare land near the roads may have been a result of roads being constructed in the study area, which affects slope stability and increases the probability of landslides. Furthermore, the bare locations near the roads slightly increased in number after the typhoon, likely because the roads changed the routes of surface runoff. The area of bare land near the streams also increased, possibly because the water flow scours the slope toe and causes continual bank

collapses. Thus, typhoons have a significant impact on the stability of slopes below mountain roads.

**7 Conclusions**

This study applied the maximum likelihood method to interpret and classify satellite images before and after two typhoons in 2011 and 2013. We extracted landslide and land use information from the areas surrounding roads and then compiled the rainfall and DEM data from the typhoon events. By using the MHEM, we established a landslide susceptibility assessment

model and examined the relationships between causal factors and the area and number of landslides within the study area, as well as the relationships between roads and the spatial distribution of landslides. The results show that the *Kappa* coefficients associated with the use of the maximum likelihood method to interpret and classify satellite images before and after Typhoon Nanmadol in 2011 and Typhoon Kong-rey in 2013 ranged from 0.53 to 0.66, whereas the *OA* ranged from 61% to 71%, indicating moderately high accuracy. According to the results of the instability index-based landslide

susceptibility assessment model, the degree of land disturbance, geology, slope gradient, and slope roughness had the greatest impacts on landslides. A comparison of historical landslides triggered by the typhoons and the results of the hazard map revealed 71% accuracy for Typhoon Nanmadol in 2011 and 75% accuracy for Typhoon Kong-rey in 2013. Regarding the influence of the causal factors, an elevation of 450–750 m, a slope gradient of 30˚–55˚, and distances within 300 m of water or roads were associated with a larger scale of landslides. The scale of landslides also increased with the degree of land

disturbance. The relationships between the ILR, instability index, and landslide occurrence indicate that for a high instability index, even a smaller rainfall event could trigger a landslide. By contrast, for a low instability index, a larger rainfall event could not easily trigger a landslide. Thus, the instability index can effectively reflect landslide susceptibility. Comparisons of the distribution of bare land before and after typhoon events showed that most landslides in the study area were caused by stream water scouring away the toes of bank slopes. Although bare locations did not significantly increase in number after

the typhoon events, they increased significantly in area, implying that the number of old landslide areas holding more collapses or expansions was greater than that of new landslide areas developing. In addition, the results obtained from observing changes in slopes above and below mountain roads after the typhoon events indicate that the number of bare





locations on the slopes above the roads in the study area did not increase significantly, whereas the bare locations near the roads on the slopes below the roads slightly increased in number after the typhoon events, likely because of the roads changing the routes of surface runoff. The amount of bare land near streams also increased, possibly because the water flow scours the slope toe.

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





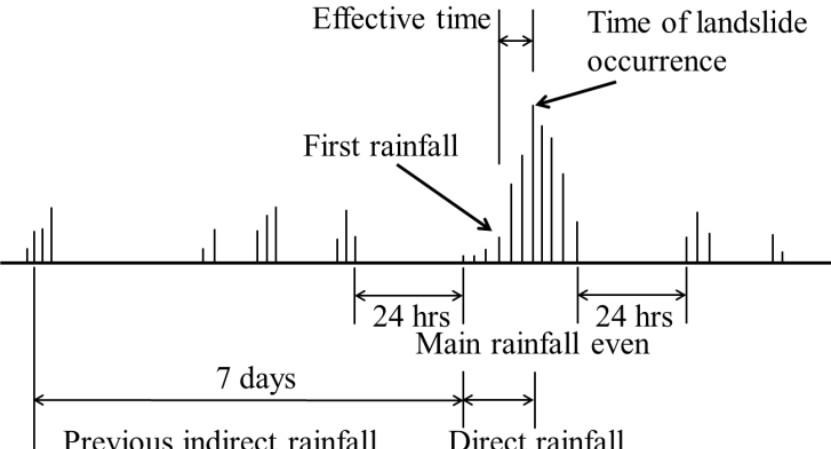

**Figure 1: Definition of Rainfall Events based on Effective Accumulated Rainfall (modified from Seo and Funasaki, 1973)**




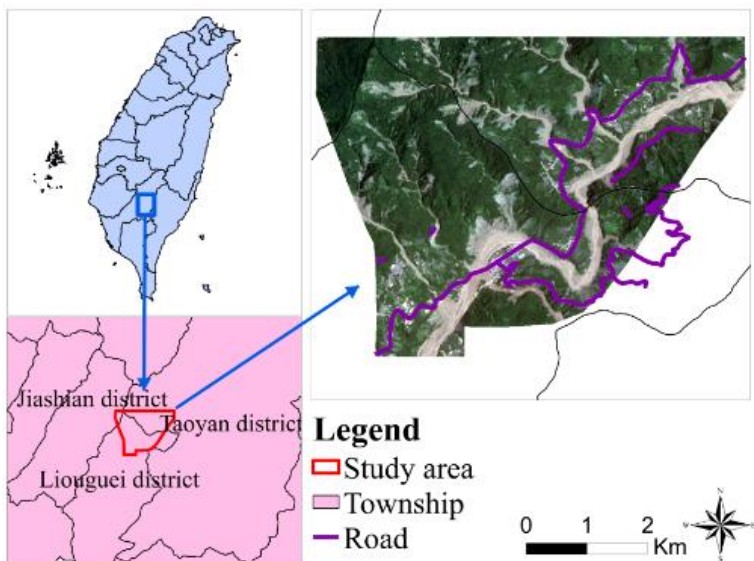

**Figure 2: Study Area in the southern Taiwan, blue line depict the distribution of mountain roads.**




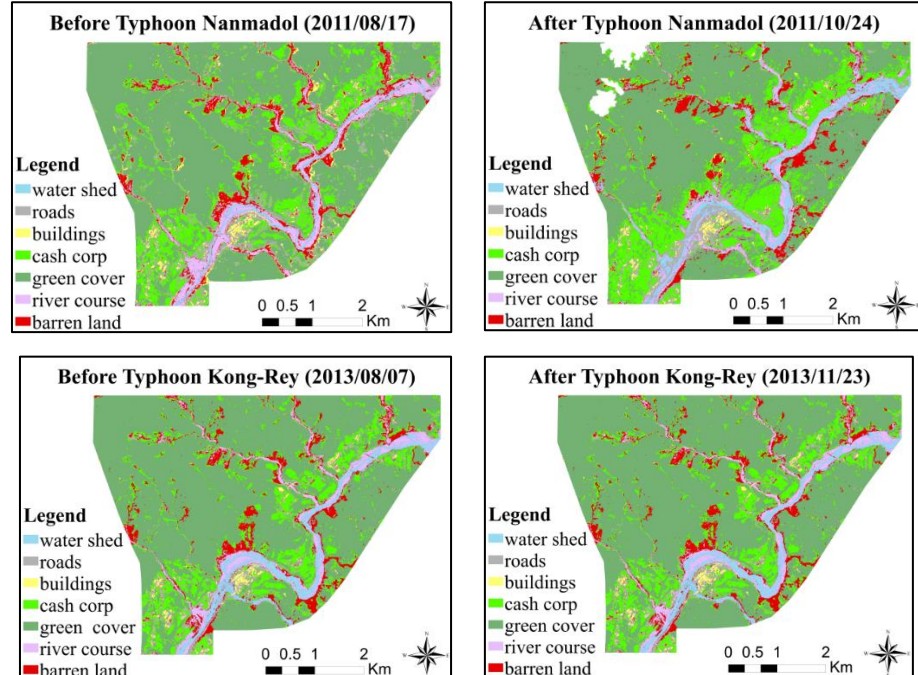

**Figure 3: Interpretation and Classification Results of Satellite Images Before (Left) and After (Right) Typhoon Nanmadol (Top) and Typhoon Kong-rey (Bottom)**



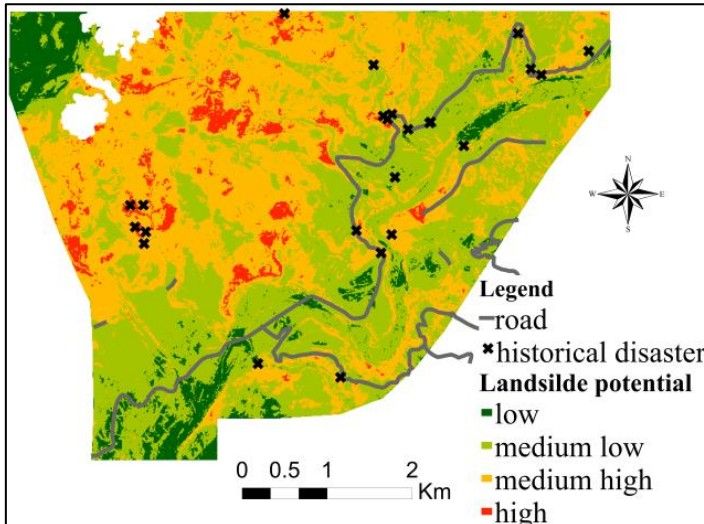

**Figure 4: Landslide Susceptibility in Study Area, in which cross symbols represent the historical disasters collected from NCDR (2017)**




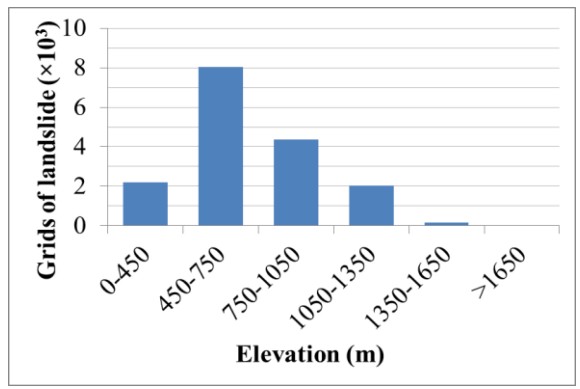

(*a*) Elevation

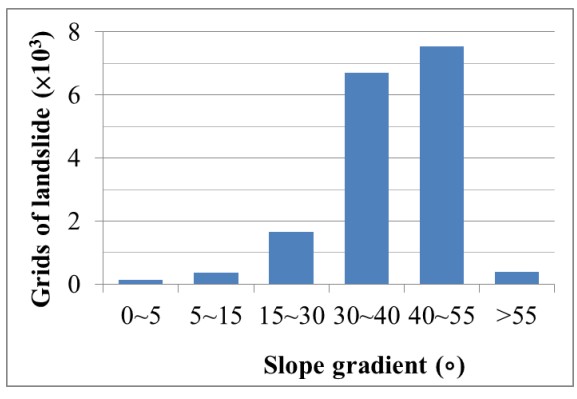

(*b*) Slope gradient

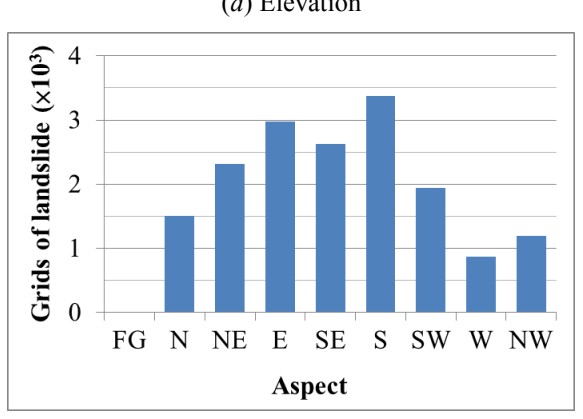

(*c*) Aspect

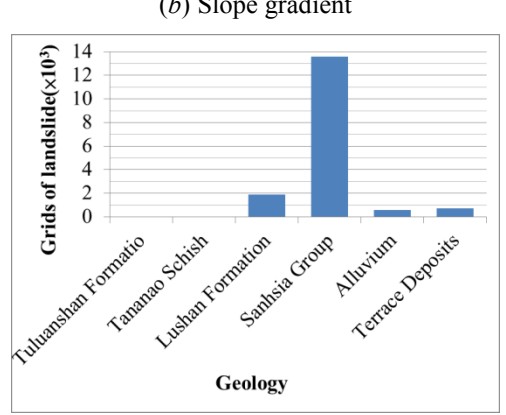

(*d*) Geology

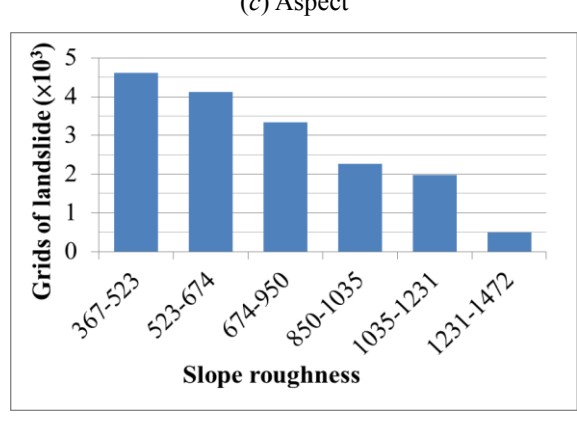

(*e*) Slope roughness

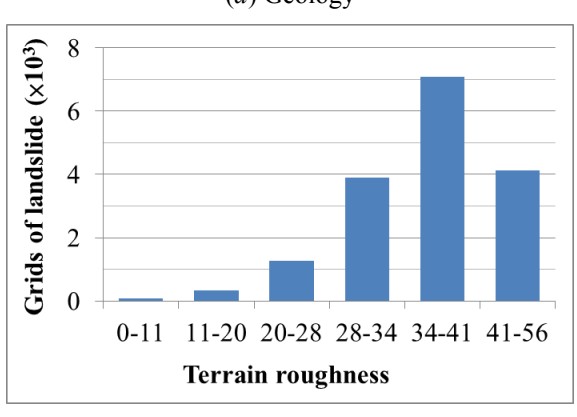

(*f*) Terrain roughness




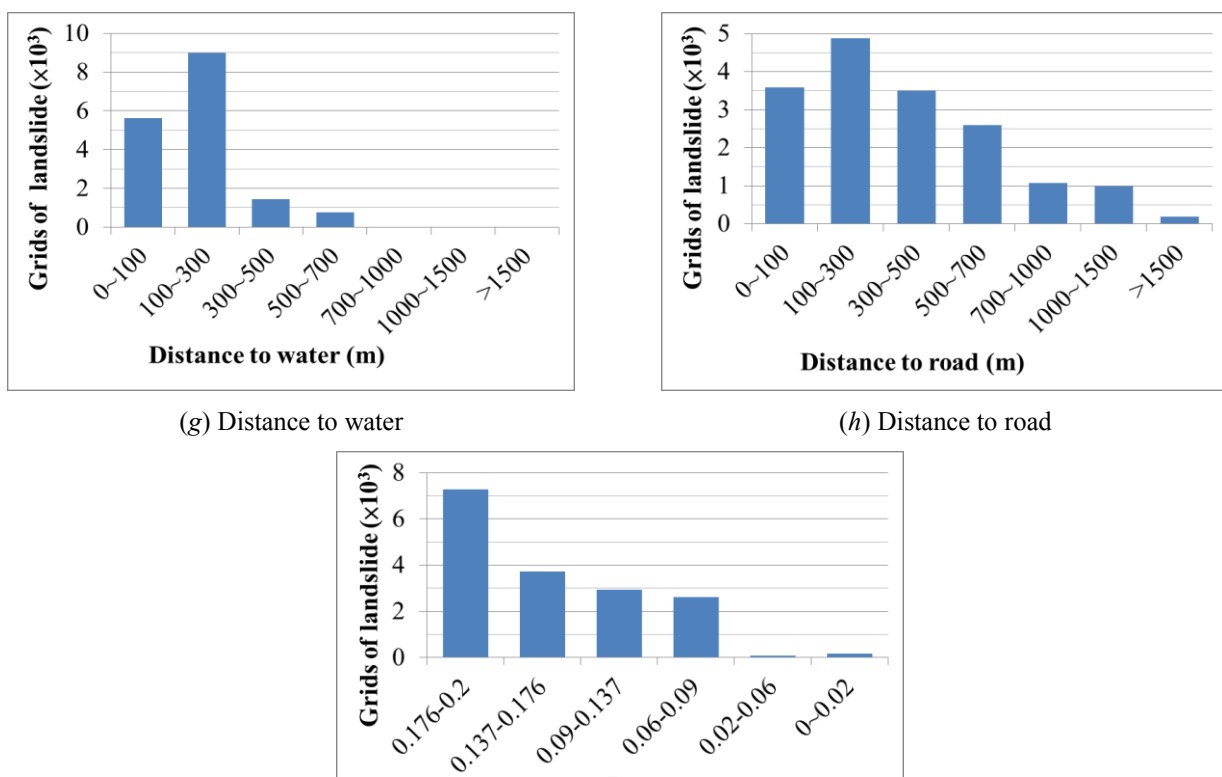

(g) Distance to water          (h) Distance to road

(i) Degree of land disturbance

**Figure 5: Relationships between Landslide Impact Factors and Number of Landslide Grids in Study Area**





(a) Typhoon Nanmadol (2011) based on *EAR*

(b) Typhoon Kong-rey (2013) based on *EAR*

(c) Typhoon Nanmadol (2011) based on $I_{3R}$

(d) Typhoon Kong-rey (2013) based on $I_{3R}$

(e) Typhoon Nanmadol (2011) based on $EAR \times I_{3R}$

(f) Typhoon Kong-rey (2013) based on $EAR \times I_{3R}$

**Figure 6: Relationships among Instability Index, Effective Accumulated Rainfall, and Landslide Occurrence in Study Area after Typhoons Nanmadol (2011) and Kong-rey (2013), respectively.**



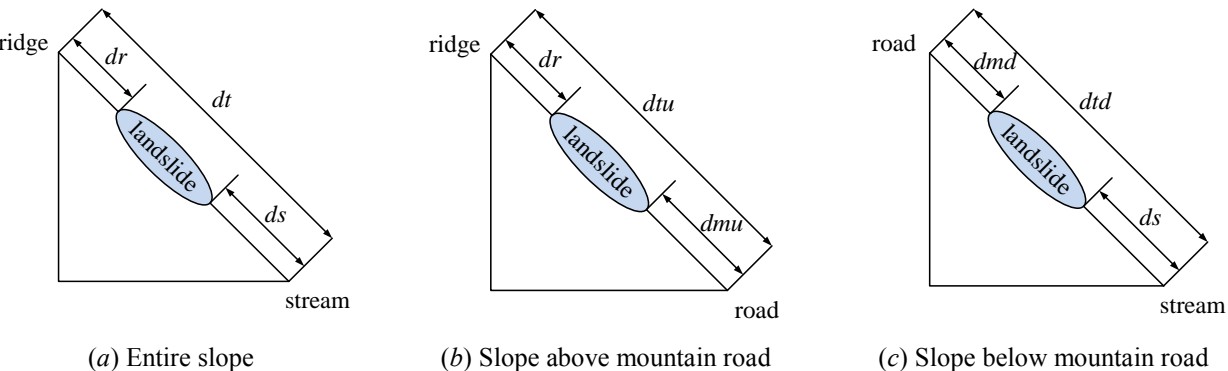

(*a*) Entire slope      (*b*) Slope above mountain road      (*c*) Slope below mountain road

**Figure 7: Diagrams of Landslide Area on Slope, in which *dr* represents the distance between the highest point of a landslide area and the nearest ridge, *ds* the distance between the lowest point of the landslide area and the nearest stream, and *dt* the distance between ridge and stream.**





**Figure 8: Spatial Distribution of Bare Land in the Study Area before and after the Typhoons Nanmadol (Top) and Typhoon Kong-rey (Bottom), the scales of bubble reflect the area of each bare land.**




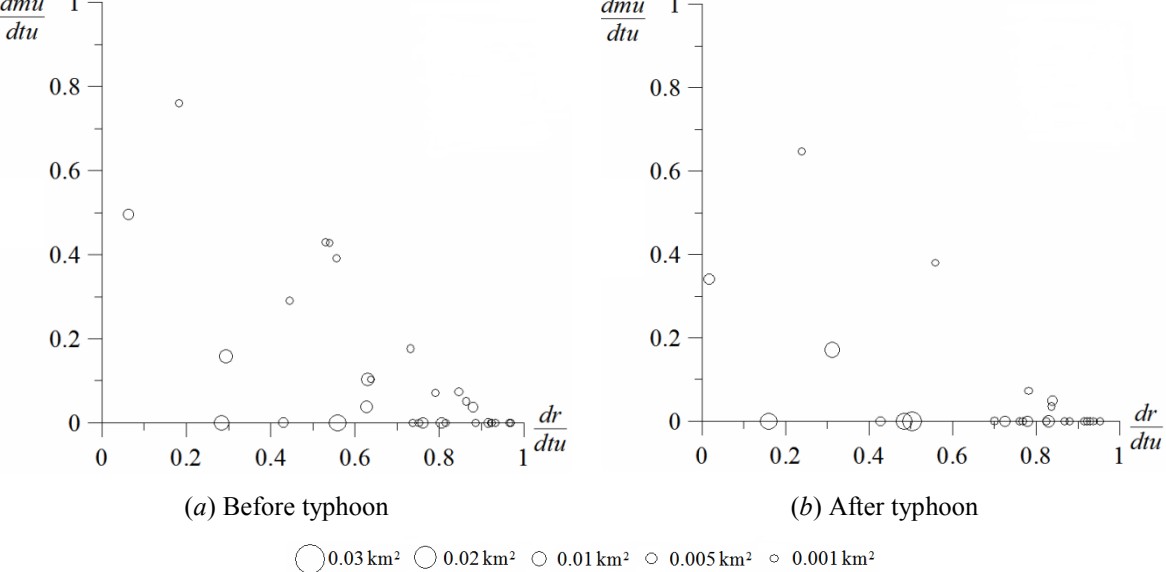

**Figure: 9 Spatial Distribution of Bare Land on Slopes above Mountain Roads in the Study Area before and after Typhoon Kong-rey in 2013, the scales of bubble reflect the area of each bare land.**





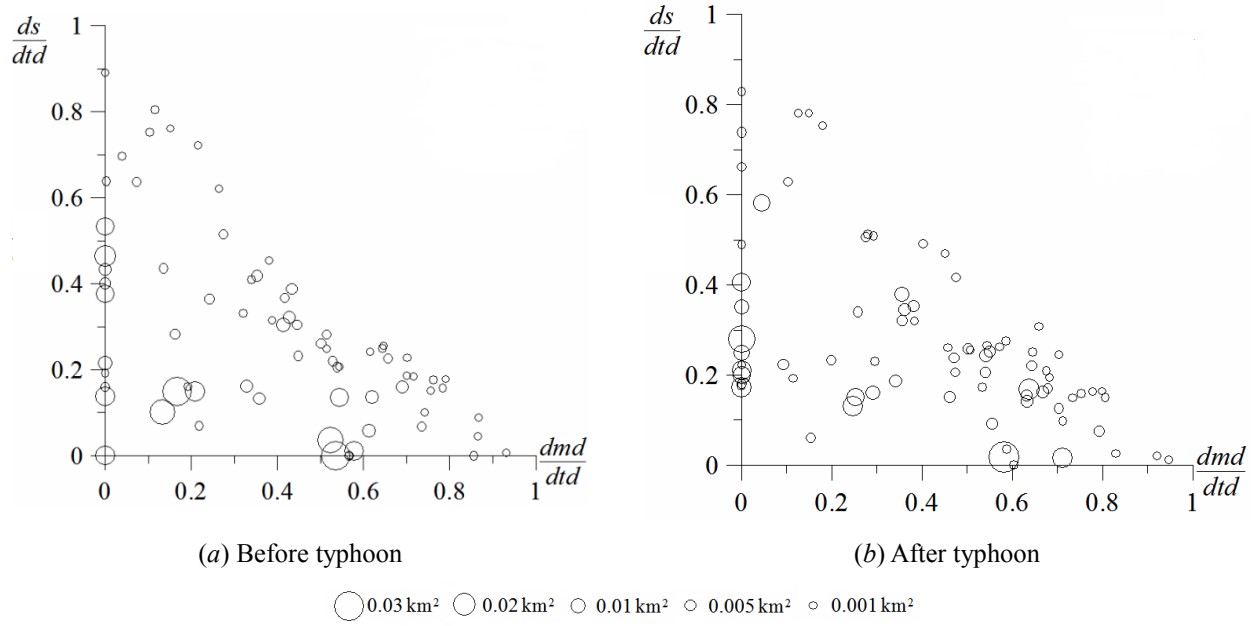

(*a*) Before typhoon          (*b*) After typhoon

○ 0.03 km²  ○ 0.02 km²  ○ 0.01 km²  ○ 0.005 km²  ○ 0.001 km²

**Figure 10: Spatial Distribution of Bare Land on Slopes below Mountain Roads in the Study Area before and after Typhoon Kong-rey in 2013, the scales of bubble reflect the area of each bare land.**



**Table 1: Relationship Table of Error Matrix (Verbyla, 1995)**

| | | Actual ground surface | | Total |
| --- | --- | --- | --- | --- |
| | | Class A | Class B | |
| Classification results | Class A | $X_{11}$ | $X_{12}$ | $X_{+i}$ |
| | Class B | $X_{21}$ | $X_{22}$ | $X_{+i}$ |
| Total | | $X_{i+}$ | $X_{i+}$ | $X_{++}$ |



**Table 2: Error Matrix of Interpretation Results of Satellite Images before and after Typhoon Kong-rey in 2013**

|  | Water | Roads | Buildings | Crops | Vegetation | River channels | Bare land | Subtotal | User's accuracy (%) |
|---|---|---|---|---|---|---|---|---|---|
| Water | 15 | 0 | 0 | 0 | 0 | 0 | 0 | 15 | 100 |
| Roads | 1 | 7 | 2 | 0 | 0 | 3 | 0 | 10 | 70 |
| Buildings | 2 | 0 | 22 | 0 | 0 | 0 | 0 | 24 | 92 |
| Crops | 0 | 4 | 0 | 11 | 0 | 0 | 1 | 16 | 69 |
| Vegetation | 1 | 5 | 0 | 12 | 25 | 0 | 2 | 45 | 56 |
| River channels | 6 | 3 | 1 | 0 | 0 | 24 | 4 | 38 | 63 |
| Bare land | 0 | 6 | 0 | 2 | 0 | 1 | 18 | 27 | 67 |
| Subtotal | 25 | 25 | 25 | 25 | 25 | 25 | 25 | 175 |  |
| Producer's accuracy (%) | 60 | 28 | 88 | 44 | 100 | 95 | 72 |  |  |
| *Kappa* = 0.64; *OA* = 70% | | | | | | | | | |





**Table 3: Interpretation Results of Satellite Images before and after Typhoons Nanmadol and Kong-rey**

| Time of satellite image | *Kappa* | *OA* (%) |
|---|---|---|
| Before Typhoon Nanmadol (2011.08.17) | 0.64 | 69 |
| After Typhoon Nanmadol (2012.10.14) | 0.53 | 61 |
| Before Typhoon Kong-rey (2013.08.17) | 0.66 | 71 |
| After Typhoon Kong-rey (2013.11.23) | 0.64 | 70 |





**Table 4: Scores of Index for Disturbance Condition (revised from Chen et al., 2009, 2013)**

| Index for disturbance condition | Bare land | Roads | Buildings | Crops | Vegetation |
|---|---|---|---|---|---|
| Score | 5 | 4 | 3 | 2 | 1 |





**Table 5: Information of weather Stations Used in This Study (Central Weather Bureau)**

| Station No. | Station name | Longitude (degrees) | Latitude (degrees) | X(TWD97 Taiwan) | Y(TWD97 Taiwan) |
|---|---|---|---|---|---|
| C1O880 | Guan-shan | 120.5941 | 23.1734 | 208443.362 | 2563542.352 |
| C0V150 | Biao-hu | 120.6647 | 23.2602 | 215693.732 | 2573135.951 |
| C1V220 | Hsiao Guan-shan | 120.8136 | 23.1542 | 230913.463 | 2561372.400 |
| C1V230 | Gao-jhong | 120.7167 | 23.1349 | 220987.130 | 2559250.525 |
| C1V240 | Sin-fa | 120.6601 | 23.0521 | 215169.331 | 2550097.989 |
| C0V250 | Jia-sian | 120.5918 | 23.0801 | 208178.971 | 2553211.279 |
| C1V270 | Xi-nan | 120.8064 | 23.0760 | 230166.772 | 2552711.336 |





**Table 6: Effective Accumulated Rainfall and Intensity of Rolling Rainfall Observed at Weather Stations during Typhoon Nanmadol and Typhoon Kong-rey**

| Weather station name | 2011 Typhoon Nanmadol | | 2013 Typhoon Kong-rey | |
|---|---|---|---|---|
| | $EAR$ | Max $I_{3R}$ | $EAR$ | Max $I_{3R}$ |
| Guan-shan | 73.79 | 57 | 376.39 | 146.5 |
| Biao-hu | 68.19 | 38.5 | 412.97 | 145 |
| Hsiao Guan-shan | 100.82 | 47.5 | 414.91 | 122.5 |
| Gaojhong | 336.97 | 68.5 | 543.85 | 135.5 |
| Sinfa | 503.94 | 61 | 288.35 | 122.5 |
| Jiasian | 378.92 | 45.5 | 233.12 | 100.5 |
| Xi-nan | 191.95 | 48 | 518.48 | 101.5 |



**Table 7: Correlation Test Results Between the Impact Factors**

|  | Elevation | Slope gradient | Aspect | Slope roughness | Terrain roughness | Distance to water | Distance to road | $I_{DC}$ | EAR |
|---|---|---|---|---|---|---|---|---|---|
| Elevation | 1 | 0.39 | -0.01 | 0.47 | 0.99 | 0.52 | 0.62 | -0.23 | -0.66 |
| Slope gradient | - | 1 | -0.07 | 0.85 | 0.37 | 0.11 | 0.18 | -0.09 | -0.19 |
| Aspect | - | - | 1 | -0.09 | -0.03 | 0.13 | 0 | 0 | -0.12 |
| Slope roughness | - | - | - | 1 | 0.48 | 0.14 | 0.23 | -0.11 | -0.25 |
| Terrain roughness | - | - | - | - | 1 | 0.53 | 0.63 | -0.24 | -0.67 |
| Distance to water | - | - | - | - | - | 1 | 0.49 | -0.21 | -0.47 |
| Distance to road | - | - | - | - | - | - | 1 | -0.14 | -0.61 |
| $I_{DC}$ | - | - | - | - | - | - | - | 1 | 0.14 |
| EAR | - | - | - | - | - | - | - | - | 1 |





**Table 8: Paired Sample t Test Results Between Elevation and Terrain Roughness and Slope Gradient and Slope Roughness**

| | Paired difference | | | | | t | Degree of freedom | Significance (Two-tailed) |
|---|---|---|---|---|---|---|---|---|
| | Mean | S.D. | Standard error of mean | 95% confidence interval of difference | | | | |
| | | | | Upper limit | Lower limit | | | |
| Elevation-terrain roughness | -2.69 | 46.5 | 0.07 | -2.83 | -2.54 | -36.8 | 407493 | 0 |
| Slope gradient-slope roughness | -0.11 | 7.9 | 0.01 | -0.14 | -0.08 | -9.1 | 407493 | 0 |



**Table 9: Weights and Scores of Causal Factors after Rainfall Brought by Typhoon Nanmadol in 2011**

| Impact factor | No. | Number of grids | Number of landslides | Landslide percentage | Score | Impact factor | No. | Number of grids | Number of landslides | Landslide percentage | Score |
|---|---|---|---|---|---|---|---|---|---|---|---|
| Elevation ($E_l$) | 1 | 0 | 0 | 0 | 1 | Slope gradient ($S_s$) | 1 | 0 | 0 | 0 | 1 |
| | 2 | 0 | 0 | 0 | 1 | | 2 | 4096 | 378 | 0.092 | 9.21 |
| | 3 | 8377 | 175 | 0.021 | 4.62 | | 3 | 74623 | 7545 | 0.101 | 10 |
| | 4 | 45633 | 2043 | 0.045 | 8.75 | | 4 | 119696 | 6704 | 0.056 | 5.99 |
| | 5 | 84049 | 4370 | 0.052 | 10 | | 5 | 100477 | 1666 | 0.017 | 2.48 |
| | 6 | 209648 | 8023 | 0.038 | 7.62 | | 6 | 61442 | 369 | 0.006 | 1.53 |
| | 7 | 59787 | 2182 | 0.036 | 7.32 | | 7 | 47160 | 131 | 0.003 | 1.25 |
| $\sigma$=0.021, $V$=0.764, $W$=0.087 | | | | | | $\sigma$=0.044, $V$=1.111, $W$=0.127 | | | | | |
| Aspect ($A_s$) | 1 | 72961 | 3381 | 0.046 | 10 | $I_{DC}$ | 1 | 18462 | 7278 | 0.394 | 10 |
| | 2 | 129113 | 4569 | 0.035 | 7.87 | | 2 | 37591 | 3735 | 0.099 | 3.26 |
| | 3 | 95534 | 3839 | 0.040 | 8.80 | | 3 | 33686 | 2924 | 0.087 | 2.97 |
| | 4 | 75666 | 3505 | 0.046 | 10 | | 4 | 78519 | 2611 | 0.033 | 1.75 |
| | 5 | 34220 | 1499 | 0.044 | 9.51 | | 5 | 216535 | 83 | 0 | 1 |
| | 6 | 0 | 0 | 0 | 1 | | 6 | 22701 | 162 | 0.007 | 1.15 |
| $\sigma$=0.018, $V$=0.504, $W$=0.058 | | | | | | $\sigma$=0.148, $V$=1.431, $W$=0.163 | | | | | |
| Slope roughness ($S_r$) | 1 | 32672 | 4136 | 0.127 | 10 | Terrain roughness ($T_r$) | 1 | 20809 | 496 | 0.024 | 1 |
| | 2 | 83465 | 7085 | 0.084 | 7.01 | | 2 | 36844 | 1969 | 0.053 | 10 |
| | 3 | 104560 | 3903 | 0.037 | 3.6 | | 3 | 47547 | 2257 | 0.047 | 8.18 |
| | 4 | 75349 | 1260 | 0.017 | 2.12 | | 4 | 67105 | 3330 | 0.050 | 8.84 |
| | 5 | 51143 | 342 | 0.007 | 1.4 | | 5 | 98836 | 4121 | 0.042 | 6.43 |
| | 6 | 60305 | 67 | 0.001 | 1 | | 6 | 136353 | 4620 | 0.034 | 4.05 |
| $\sigma$=0.05, $V$=1.098, $W$=0.125 | | | | | | $\sigma$=0.011, $V$=0.266, $W$=0.03 | | | | | |
| Distance to water ($D_s$) | 1 | 134641 | 5610 | 0.042 | 8.08 | Distance to road ($D_r$) | 1 | 165766 | 3581 | 0.022 | 1 |
| | 2 | 169659 | 8983 | 0.053 | 10 | | 2 | 120008 | 4871 | 0.041 | 3.08 |
| | 3 | 69076 | 1446 | 0.021 | 4.56 | | 3 | 44993 | 3505 | 0.078 | 7.16 |
| | 4 | 19906 | 754 | 0.038 | 7.44 | | 4 | 25015 | 2597 | 0.104 | 10 |
| | 5 | 8336 | 0 | 0 | 1 | | 5 | 25101 | 1065 | 0.042 | 3.28 |
| | 6 | 5627 | 0 | 0 | 1 | | 6 | 21848 | 986 | 0.045 | 3.58 |
| | 7 | 249 | 0 | 0 | 1 | | 7 | 4763 | 188 | 0.039 | 2.96 |



| Impact factor | No. | Number of grids | Number of landslides | Landslide percentage | Score | Impact factor | No. | Number of grids | Number of landslides | Landslide percentage | Score |
|---|---|---|---|---|---|---|---|---|---|---|---|
| | | *σ*=0.023, *V*=1.029, *W*=0.117 | | | | | | *σ*=0.028, *V*=0.528, *W*=0.058 | | | |
| *EAR* | 1 | 15768 | 139 | 0.00882 | 2.05196 | Geology ($G_s$) | 1 | 70071 | 738 | 0.011 | 2.56 |
| | 2 | 113386 | 3590 | 0.03166 | 4.77831 | | 2 | 43675 | 598 | 0.014 | 3.02 |
| | 3 | 163395 | 7879 | 0.04822 | 6.75433 | | 3 | 222814 | 13575 | 0.061 | 10 |
| | 4 | 73522 | 3191 | 0.0434 | 6.17931 | | 4 | 70934 | 1882 | 0.027 | 4.92 |
| | 5 | 26439 | 1994 | 0.07542 | 10 | | 5 | 0 | 0 | 0 | 1 |
| | 6 | 14984 | 0 | 0 | 1 | | 6 | 0 | 0 | 0 | 1 |
| | | *σ*=0.028, *V*=0.797, *W*=0.091 | | | | | | *σ*=0.023, *V*=1.233, *W*=0.141 | | | |



**Table 10: Intervals of Instability Index and Landslide Probability of Rainfall Factors**

| Rainfall factor | $D_{t,min}$ | $D_{t,max}$ | $P(F)_{min}$ | $P(F)_{max}$ |
|---|---|---|---|---|
| $EAR$ | 2.05 | 9.59 | 0.312 | 0.982 |
| $I_{3R}$ | 2.02 | 9.96 | 0.305 | 0.998 |





**Table 11: Accuracy of Landslide Susceptibility Map in Considering Different Rainfall Factors and Typhoons**

| Typhoon event | Rainfall factor | Landslide susceptibility at locations of 24 historical disasters | | | | Accuracy (%) | Mean accuracy (%) |
|---|---|---|---|---|---|---|---|
| | | Low susceptibility | Medium low susceptibility | Medium high susceptibility | High susceptibility | | |
| Typhoon Nanmadol (2011) | $EAR$ | 2 | 5 | 11 | 6 | 71% | 71% |
| | $I_{3R}$ | 3 | 4 | 13 | 4 | 71% | |
| Typhoon Kong-rey (2013) | $EAR$ | 2 | 4 | 13 | 5 | 75% | 75% |
| | $I_{3R}$ | 2 | 4 | 11 | 7 | 75% | |



**Table 12: Numbers of Landslide Grids in Study are corresponding to Different $D_t$ Levels under Different Rainfall Factors after Typhoons**

| Rainfall event | Numbers of landslide and non-landslide grids (Proportion of landslide grids) | | | Number of grids in each level based on $EAR$ | | | Number of grids in each level based on $I_{3R}$ | | |
|---|---|---|---|---|---|---|---|---|---|
| | | | | $D_t$ level | | | $D_t$ level | | |
| | | | | Low | Medium | High | Low | Medium | High |
| Typhoon Nanmadol (2011) | Whole area | Landslide | 16793 | 211 | 3031 | 13551 | 216 | 3603 | 12974 |
| | | Non-landslide | 390710 | 168259 | 166289 | 56153 | 177396 | 166358 | 46947 |
| | | (Landslide/ Non-landslide) | | 0.00125 | 0.01822 | 0.24132 | 0.00122 | 0.02166 | 0.27635 |
| | Random sampling | Landslide | 50 | 0 | 11 | 39 | 0 | 12 | 38 |
| | | Non-landslide | 50 | 24 | 21 | 5 | 25 | 21 | 4 |
| Typhoon Kong-rey (2013) | Whole area | Landslide | 20771 | 392 | 4303 | 16076 | 434 | 4482 | 15855 |
| | | Non-landslide | 396175 | 182810 | 181824 | 31541 | 181079 | 185305 | 29791 |
| | | (Landslide/ Non-landslide) | | 0.00214 | 0.02367 | 0.50969 | 0.00240 | 0.02419 | 0.53221 |
| | Random sampling | Landslide | 50 | 1 | 6 | 43 | 0 | 11 | 39 |
| | | Non-landslide | 50 | 27 | 20 | 3 | 20 | 27 | 3 |