# Peer review of "Scale and spatial distribution assessment of rainfall-induced landslides in a catchment with mountain roads"

_Natural Hazards and Earth System Sciences, 2017_

## Referee Comment (RC1) · Anonymous Referee #1 · 1 Nov 2017

General comments The paper proposes an analysis of landslide susceptibility in a mountain area, crossed by a road and affected by landslides triggered by typhoons. The topic could be interesting to NHESS readers, if some issues are more clearly presented, in particular, the aims, the used methods (in a right temporal sequence), and expected results. For this reasons, a major revision is needed before its being accepted for publication.

Specific comments Please define clearly what is the aim of scale assessment. The terminology should be checked and made uniform, with reference to the following terms: causal factors, predisposing factors, impact factors, landslide-inducing factors. Refer-

ence description is not well presented, sometimes redundant, sometimes limited. Too repetition of "studies, many studies, previous studies, several studies, early research". I suggest to discuss methods and procedures available in literature, avoiding to refer to single reference with expression as, for example, X et al. used [. . .], Y et al. described [. . .], Z et al. utilized [. . .]. The introduction and especially the literature discussion (pages 2 and 3) about the landslide susceptibility assessment methods must be re-organized and rewritten using a clearly and well-ordered structure. Aims, procedures and expected results are not clearly defined either in introduction either in methodological section. It is not clear if the study are is only the highway or the whole catchment crossed by road; the study area seems to be the road according to title, but the final susceptibility map, in figure 4, is referred to the whole area. So the presence of the road is negligible at the aim of the analysis. The title does not reflect clearly the contents of the paper. Please, rephrase the paragraph 3, adding more information and details about the study area. About methodology, does maximum likelihood method have any disadvantages? Was the error associated with this automatic image interpretation technique calculated? Please rewrite the paragraph 2.2 in order to describe more clearly the MHEM method. I suggest to reconsider the title, because the analysis was not performed only along the road but in the surrounding territory and the image interpretation does not emerge from the title. It is not completely coherent with the contents of the paper. Please, reorganize the paper, by separating the description of methodology from the discussion of results. There are too much paragraphs that make confusing and difficult the readability and understanding of performed analyses, in particular from paragraph 5.2 onwards.

Technical corrections Pag 1 line 8: please, move "Typhoons" at the end of the sentence. Pag 1 line 10: "topographic changes" or "surface changes" instead of "changes in slope surface". Pag 1 line 10: "A multivariate statistical method" instead of "The multivariate hazard evaluation method". Pag. 1 lines 11-12: Please, rephrase the sentence. The evaluation of landslide locations and relationship between landslide and predisposing factors is preparatory for assessing and mapping landslide susceptibility. Pag. 1 line

26: please, replace "occurrence distribution" with "distribution of existing landslides" and "a set of predisposing factors such as geo-environmental thematic variables" with "a set of geo-environmental predisposing factors". Pag. 1 line 27: "sediment disaster" is not an appropriate expression; please, replace it with landslides. Pag. 2 lines 1-3: Please modify the terminology used in this sentence. "predisposing factors" instead of "potential causes" and "triggering factors" instead of "impetuses". Pag. 2 lines 7-8: This sentence is a repetition. Pag. 2 lines 9-10:please add references about model uncertainty evaluation, for example: Wang X, Frattini P, Crosta GB, Zhang L, Agliardi F, Lari S, Yang Z. 2014. Uncertainty assessment in quantitative rockfall risk assessment. Landslides. 11:711–722. Pag. 2 line 10: explain what is the meaning of scale in this study: size, intensity of landslide? Pag. 2 line 11: This sentence is a repetition. Pag. 2 lines 12-13: The meaning of this sentence is unclear. Pag. 2 lines 14-16: This sentence is a repetition. Pag. 2: I suggest to add some reference about AHP method (1), multivariate statistical methods (2) and landslide susceptibility assessment along roads (3): (1) Kayastha P., Dhital M.R., De Smedt F. 2013. Application of the analytical hierarchy process (AHP) for landslide susceptibility mapping: A case study from the Tinau watershed, west Nepal. Computers Geosciences, 52: 398-408 (1) Zhang G., Cai Y., Zheng Z., Zhen J., Liu Y., Huang K. 2016. Integration of the Statistical Index Method and the Analytic Hierarchy Process technique for the assessment of landslide susceptibility in Huizhou, China. CATENA, 142: 233-244. (2) Carrara A, Crosta G, Frattini P. 2008. Comparing models of debris-flow susceptibility in the alpine environmental. Geomorphology. 94:353–378. (2) Pellicani R, Frattini P, Spilotro G. 2014. Landslide susceptibility assessment in Apulian Southern Apennine: heuristic vs. statistical methods. Environ Earth Sci. 72:1097–1108. doi: 10.1007/s12665-013-3026-3 (3) Pellicani R, Spilotro G, Van Westen CJ. 2016. Rockfall trajectory modelling combined with heuristic analysis for assessing the rockfall hazard along the Maratea SS18 coastal road (Basilicata, southern Italy). Landslides. 13:985–1003. (3) Pantelidis L. 2011. A critical review of highway slope instability risk assessment systems. Bull Eng Geol Environ. 70:395–400. (3) Devkota KC, Regmi AD, Pourghasemi HR,

Yoshida K, Pradhan B. 2013. Landslide susceptibility mapping using certainty factor, index of entropy and logistic regression models in GIS and their comparison at Mugling–Narayanghat road section in Nepal Himalaya. Nat Hazards. 65:135–165. doi: 10.1007/s11069-012-0347-6 (3) Pellicani R., Argentiero I., Spilotro G. (2017) GIS-based predictive models for regional-scale landslide susceptibility assessment and risk mapping along road corridors. Geomatics, Natural Hazards and Risk, 1-22. DOI: 10.1080/19475705.2017.1292411. Pag. 3 line 34: It is not clear how and from where the location of landslides was extracted? Are existing or potential landslides? The evaluation of landslide locations and the relationship between landslides and predisposing factors is preparatory for assessing and mapping landslide susceptibility. Pag.5 line 13: please replace "risk" with "susceptibility". Pag. 5 lines 14-15: Please avoid repetitions: variability, variance. Pag. 5 lines 18-19: Please rewrite this sentence using a correct terminology, "cell" or "pixel" instead of "grid" and "class" instead of "grade". Pag. 5 lines 22-23: Please rewrite this sentence, a confusing terminology has been used (causal factor, impact factor, grades). Pag.6: Which is the difference between factor weight and graded score? It is not clear. Pag. 8 line 13: why 0.9? Pag. 8 line 17: Is EAR expressed in mm? Pag. 8 line 21: Is Ir expressed in mm/h? Pag.8 line 26: What is the meaning of rolling hours? Pag. 10 line 6:" thematic map of predisposing factors" instead of " map of the natural environment". Pag. 10 line 9: please make uniform the terminology, as for example causal factors, predisposing factors, impact factors, landslide-inducing factors, etc. Pag.10 lines 21-22-26: please, modify "grid" and "grades". Pag. 10 line 27: explicit the values of the six categories. Pag. 11 line 4: what is the meaning of "geological strength"? The geological map should be classified into classes corresponding to different formations or lithological units. Pag. 11 line 9: define the analysis function. Pag. 11 lines 10 and 16: explicit the six classes. Pag. 11 lines 18 and 20: give more information about two factors. Pag. 11 line 21: Land disturbance looks like a reclassified land use map. The highest score of disturbance is assigned to bare land, why not to roads and buildings? This is a qualitative attribution, it should be written somewhere.

---

## Referee Comment (RC2) · Anonymous Referee #2 · 30 Nov 2017

General comments: The authors propose in this paper an assessement of landslide susceptibility in a mountain area in Taiwan. The manuscript, which can be interesting for people studing relations between landslide susceptibility and hydrology, has several problems that can be improved after a minor revision. Readers more interested with interactions between natural hazards and roads stay more on the sidelines.

Specifics comments: The state of the art of the methods to evaluate factors influencing landslides in the Introduction is well detailed but can be better structured. I suggest to add more information / specifications about the study area (surface, length, meters above and under the road path, etc.) and the road (type, traffic, closure consequences,

length, history, etc.). The presence of the "road" term in the title does not well represent the manuscript content. It should be more focused about the road. Please define and describe the "landslide" term used in this paper (area, volume, depth, geology, etc.). What kind of landslides do you consider? There are to much subchapters (2.3.1, 2.3.2, etc.), to much figures and tables in my opinion. I suggest to move some of them in appendices (as Table 9 for instance). Please try to reduce the number of subchapters and keep only the really relevant Figures and Tables for the comprehension of the manuscript.

Technical corrections: - Page 1, line 18 : mm value for the annually rainfall is wrong, it should be: 2'506 mm and not 2.506 mm. - Page 2, line 23 : Are really images consistent in quality ? Clouds, shadows, etc. - Page 3, line 23 : Studies have indicated... Which studies ? - Page 3, line 1 : The results of this study could serve as a reference.. Maybe too presumptuous. - Page 4, line 18 : Please define x. - Page 5, line 5 : Please define k. - Page 6, line 9 : Please define d. - Page 6, line 25 : In the table: Please add the Table number. - Page 7, line 5 : Please give the complete name of OA. - Page 7, line 15 : Please give explanation of Xi+, X+i, Xii. -Page 7, line 23 : Please give the complete name of EAR. -Page 8, line 20 : Please give the complete name of IR. -Page 8, line 25 and 26 : contradiction between I (rainfall intensity) and IR (not explained). -Page 9, line 14-15 : Please reword the sentence. -Page 9, 4.1, please give image info's (resolution, surface, etc.). -Page 9, line 25 : different interpretation factors: which ones ? -Page 10, line 4 : why 8 x 8 m (and not 10 x 10 m or 5 x 5 m) ? -Page 10, line 5 : we also constructed an ... DEM: how ? -Page 10, line 16 : 1480.6 and 265.2 m: are the values after the dot really needed? -Page 10, line 23 : seven grades: why seven, for what reason ? -Page 10, line 22 : seven grades: why seven, for what reason ? -Page 10, line 27 : six categories: why six, for what reason ? -Page 11, line 5 : five grades: why five, for what reason ? -Page 11, line 10 : six grades: why six, for what reason ? -Page 11, line 16 : six grades: why six, for what reason ? -Page 11, line 18 : seven grades: why seven, for what reason ? -Page 11, line 20 : seven grades: why seven, for what reason ? -Page 12, line 2 : six grades: why six, for what reason ?

-Page 12, line 7 : SPSS : maybe add Âń software Âż to more better describe what it is.
-Page 12, line 25 : seven grades: why seven, for what reason ? -Page 13, lines 3 and
4 : please clarify the sentence with the values in " () " : 2.02 and 9.96 = I3R. -Page 13,
line 20 : four level : why this repartition and not 0-25, 25-50, 50-75 and 75-100 ? All
figure and table captions : please verify that every caption is ended by a ".". - Page 24,
caption Figure 2 : which "blue line" do you mean ? Please try to redo the image (for
example the "t" Toayan district, is not well readable), colours are no well appropriated.
- Page 35, Table 2 : where aere the "before" and "after" data in the error matrix (lines
or columns) ? Please clarify. - Page 38 : Table 5 : table not necessary / relevant for
the paper. -Page 39 : Table 6 : please give units. -Page 44 : Table 10 : please define
Dt,min and Dt,max.

---

## Author Response (AR1)

**Response to reviewer comments**

on the manuscript no.: nhess-2017-264

**Scale and spatial distribution assessment of rainfall-induced landslides along mountain roads**

revised for publication in

Natural Hazards and Earth System Sciences

by

Chih-Ming Tseng, Yie-Ruey Chen, Szu-Mi Wu

First of all, we wish to thank the two reviewers for their valuable comments to the manuscript and constructive suggestions that significantly improved the manuscript. In this revised version of the paper, we have tried our best to address the comments and incorporate as much of reviewers' recommendations. Our detailed reply to two reviewers' comments are reported below. Except this response, we prepared two manuscript files, one with "track changes", another is clean version.

**Reviewer 1**

**General comments**

C1: The paper proposes an analysis of landslide susceptibility in a mountain area, crossed by a road and affected by landslides triggered by typhoons. The topic could be interesting to NHESS readers, if some issues are more clearly presented, in particular, the aims, the used methods (in a right temporal sequence), and expected results.

R1: We have confirmed the reviewer's comments and the aims, the used methods and results have been reorganized in the revised manuscript. Detailed responses are listed in specific comments and technical corrections, respectively.

**Specific comments**

C2: Please define clearly what is the aim of scale assessment.

R2: The scale of landslide in our study is defined as landslide area. The scale assessment aims to understand the relationship between the natural environment and the spatial distribution of the landslide areas. Related descriptions are also supplemented in the fourth paragraphs of introduction, as shown in P.3, Ln.15-18 of the revised manuscript.

C3: The terminology should be checked and made uniform, with reference to the following terms: causal factors, predisposing factors, impact factors, landslide-inducing factors.

R3: Done, the term "predisposing factors" was used throughout in the revised manuscript for consistency (P.6, Ln.3, Ln.5; P.7, Ln.13, Ln.16, Ln.23; P.8, Ln.3, Ln.4, Ln.5, Ln.9; P.11, Ln.1, Ln.2, Ln3; P.13, Ln.3, Ln.10; P.15, Ln.8, Ln.9, Ln.10, Ln.11; P.18, Ln.5, Ln.13).

C4: Reference description is not well presented, sometimes redundant, sometimes limited. Too repetition of "studies, many studies, previous studies, several studies, early research". I suggest to discuss methods and procedures available in literature, avoiding to refer to single reference with expression as, for example, X et al. used […], Y et al. described […], Z et al. utilized […]. The introduction and especially the literature discussion (pages 2 and 3) about the landslide susceptibility assessment methods must be reorganized and rewritten using a clearly and well-ordered structure.

R4: Done, the literature review in the Introduction has been entirely reorganized and rewritten, we follow reviewer's suggestions to use a clearly and well-ordered structure to demonstrate related references, as shown in P.1, Ln.25 - P.2, Ln.30 of the revised manuscript.

C5: Aims, procedures and expected results are not clearly defined either in introduction either in methodological section. It is not clear if the study is only the highway or the whole catchment crossed by road; the study area seems to be the road according to title, but the final susceptibility map, in figure 4, is referred to the whole area. So the presence of the road is negligible at the aim of the analysis. The title does not reflect clearly the contents of the paper.

R5: We have changed the title of this paper to "Scale and spatial distribution assessment of rainfall-induced landslides in a catchment with mountain roads" to reflect the contents of the paper.

C6: Please, rephrase the paragraph 3, adding more information and details about the study area.

R6: Done, we added more information like climate, rainfall conditions and road information of study area in the revised manuscript (P.9, Ln.12-21).

C7: About methodology, does maximum likelihood method have any disadvantages? Was the error associated with this automatic image interpretation technique calculated?

R7: The maximum likelihood method may be unsatisfactory for data with a non-normal distribution (Otukei and Blaschke, 2010), and could possible cause part error of automatic image interpretation.

Otukei, J. R. and Blaschke, T.: Land cover change assessment using decision trees, support vector machines and maximum likelihood classification algorithms, International Journal of Applied Earth Observation and Geoinformation, 12(1), S27-S31, doi: 10.1016/j.jag.2009.11.002, 2010.

C8: Please rewrite the paragraph 2.2 in order to describe more clearly the MHEM method.

R8: Done, we supplemented more descriptions of the MHEM method (paragraph 2.4 in the revised manuscript), as shown in P.7, Ln.13-14, Ln.23-24 of the revised manuscript.

C9: I suggest to reconsider the title, because the analysis was not performed only along the road but in the surrounding territory and the image interpretation does not emerge from the title. It is not completely coherent with the contents of the paper.

R9: Done, we have changed the title of this paper.

C10: Please, reorganize the paper, by separating the description of methodology from the discussion of results. There are too much paragraphs that make confusing and difficult the readability and understanding of performed analyses, in particular from paragraph 5.2 onwards.

R10: Done, we moved the calculation formula of instability index and probability from section 5.2 to the methodology section 2.4, as shown in P.8, Ln.22 - P.9, Ln.6 of the revised manuscript. And the methods were rearranged in a right temporal sequence (2.1 Maximum likelihood; 2.2 Accuracy assessment; 2.3 Rainfall analysis method; 2.4 MHEM). For the results, we moved original section 6.2 "Investigation of rainfall factors and instability index" to section 5.3 for better readability.

**Technical corrections**

C11: Page 1 line 8: please, move "Typhoons" at the end of the sentence.

R11: Done, we have moved "Typhoons" to the end of the sentence (P.1, Ln.9).

C12: Page 1 line 10: "topographic changes" or "surface changes" instead of "changes in slope surface".

R12: Done, "changes in slope surface" was replaced by "surface changes", as shown in P.1, Ln.10 of the revised manuscript.

C13: Page 1 line 10: "A multivariate statistical method" instead of "The multivariate hazard evaluation method".

R13: Done, the sentence has been modified in the revised manuscript (P.1, Ln.10).

C14: Page 1 lines 11-12: Please, rephrase the sentence. The evaluation of landslide locations and relationship between landslide and predisposing factors is preparatory for assessing and mapping landslide susceptibility.

R14: Done, the sentence was rephrased in the revised manuscript (P.1, Ln.11-12).

C15: Page 1 line 26: please, replace "occurrence distribution" with "distribution of existing landslides" and "a set of predisposing factors such as geo-environmental thematic variables" with "a set of geo-environmental predisposing factors".

R15: The literature review in the introduction has been entirely reorganized and rewritten, as shown in P.1, Ln.25 - P.2, Ln.30 of the revised manuscript.

C16: Page 1 line 27: "sediment disaster" is not an appropriate expression; please, replace it with landslides.

R16: The literature review in the introduction has been entirely reorganized and rewritten, as shown in P.1, Ln.25 - P.2, Ln.30 of the revised manuscript.

C17: Page 2 lines 1-3: Please modify the terminology used in this sentence. "predisposing factors" instead of "potential causes" and "triggering factors" instead of "impetuses".

R17: Done, we changed the term usage in the revised manuscript (P.6, Ln.3, Ln.5; P.7, Ln.13, Ln.16, Ln.23; P.8, Ln.3, Ln.4, Ln.5, Ln.9; P.11, Ln.1, Ln.2, Ln3; P.13, Ln.3, Ln.10; P.15, Ln.8, Ln.9, Ln.10, Ln.11; P.18, Ln.5, Ln.13).

C18: Page 2 lines 7-8: This sentence is a repetition.

R18: The literature review in the introduction has been entirely reorganized and rewritten, as shown in P.1, Ln.25 - P.2, Ln.30 of the revised manuscript.

C19: Page 2 lines 9-10:please add references about model uncertainty evaluation, for example: Wang X, Frattini P, Crosta GB, Zhang L, Agliardi F, Lari S, Yang Z. 2014. Uncertainty assessment in quantitative rockfall risk assessment. Landslides. 11:711–722.

R19: The literature review in the introduction has been entirely reorganized and rewritten, as shown in P.1, Ln.25 - P.2, Ln.30 of the revised manuscript.

C20: Page 2 line 10: explain what is the meaning of scale in this study: size, intensity of landslide?

R20: The scale of landslide in our study is defined as landslide area.

C21: Page 2 line 11: This sentence is a repetition.

R21: The literature review in the introduction has been entirely reorganized and rewritten, as shown in P.1, Ln.25 - P.2, Ln.30 of the revised manuscript.

C22: Page 2 lines 12-13: The meaning of this sentence is unclear.

R22: The literature review in the introduction has been entirely reorganized and rewritten, as shown in P.1, Ln.25 - P.2, Ln.30 of the revised manuscript.

C23: Page 2 lines 14-16: This sentence is a repetition.

R23: The literature review in the introduction has been entirely reorganized and rewritten, as shown in P.1, Ln.25 - P.2, Ln.30 of the revised manuscript.

C24: Page 2: I suggest to add some reference about AHP method (1), multivariate statistical methods (2) and landslide susceptibility assessment along roads (3): (1) Kayastha P., Dhital M.R., De Smedt F. 2013. Application of the analytical hierarchy process (AHP) for landslide susceptibility mapping: A case study from the Tinau watershed, west Nepal. Computers Geosciences, 52: 398-408 (1) Zhang G., Cai Y., Zheng Z., Zhen J., Liu Y., Huang K. 2016. Integration of the Statistical Index Method and the Analytic Hierarchy Process technique for the assessment of landslide susceptibility in Huizhou, China. CATENA, 142: 233-244. (2) Carrara A, Crosta G, Frattini P. 2008. Comparing models of debris-flow susceptibility in the alpine environmental. Geomorphology. 94:353–378. (2) Pellicani R, Frattini P, Spilotro G. 2014. Landslide susceptibility assessment in Apulian Southern Apennine: heuristic vs. statistical methods. Environ Earth Sci. 72:1097–1108. doi: 10.1007/s12665-013-3026-3 (3) Pellicani R, Spilotro G, Van Westen CJ. 2016. Rockfall trajectory modelling combined with heuristic analysis for assessing the rockfall hazard along the Maratea SS18 coastal road (Basilicata, southern Italy). Landslides. 13:985–1003. (3) Pantelidis L. 2011. A critical review of highway slope instability risk assessment systems. Bull Eng Geol Environ. 70:395–400. (3) Devkota KC, Regmi AD, Pourghasemi HR, Yoshida K, Pradhan B. 2013. Landslide susceptibility mapping using certainty factor, index of entropy and logistic regression models in GIS and their comparison at Mugling–Narayanghat road section in Nepal Himalaya. Nat Hazards. 65:135–165. doi: 10.1007/s11069-012-0347-6 (3) Pellicani R., Argentiero I., Spilotro G. (2017) GIS-based predictive models for regional-scale landslide susceptibility assessment and risk mapping along road corridors. Geomatics, Natural Hazards and Risk, 1-22. DOI: 10.1080/19475705.2017.1292411.

R24: Done, the literature review in the introduction has been entirely rewritten and incorporated all the references suggested by reviewer, as shown in P.1, Ln.25 - P.2, Ln.30 of the revised manuscript.

C25: Page 3 line 34: It is not clear how and from where the location of landslides was extracted? Are existing or potential landslides?

R25: The location of landslides was extracted from existing landslides of satellite images and verified by some field surveys.

C26: Page 5 line 13: please replace "risk" with "susceptibility".

R26: Done, we replaced "risk" with "susceptibility", as shown in P.7, Ln.12 of the revised manuscript.

C27: Page 5 lines 14-15: Please avoid repetitions: variability, variance.

R27: Done, we simplified the description in the revised manuscript (P.7, Ln.12-13).

C28: Page 5 lines 18-19: Please rewrite this sentence using a correct terminology, "cell" or "pixel" instead of "grid" and "class" instead of "grade".

R28: Done, the terminology has been replaced by reviewer's suggestions in the revised manuscript (P.7, Ln.19, Ln.20, Ln.22; P.8, Ln.4; Ln.5, Ln.6, Ln.8, Ln.12; P.11, Ln.10-11, Ln.14, Ln.15, Ln.19, Ln.26, Ln.28; P.12, Ln.2, Ln.4, Ln.8, Ln.11, Ln.14, Ln.20, Ln.21, Ln.28, Ln.29; P.13, Ln.17, Ln.18-19, Ln.20, Ln.23-24; P.14, Ln.22, Ln.23, Ln.25; P.15, Ln.10-11, Ln.13, Ln.14, ln.15, Ln,16, Ln.17-18, Ln.19, Ln.20-21, Ln.22, Ln.26; P.16, Ln.3, Ln.6, Ln.7-8, Ln,10, Ln.11, Ln.15, Ln.16, Ln.22, Ln.23).

C29: Page 5 lines 22-23: Please rewrite this sentence, a confusing terminology has been used (causal factor, impact factor, grades).

R29: Done, the term "predisposing factors" was used throughout in the revised manuscript for the consistency and the term "grades" was replaced by "classes".

C30: Page 6: Which is the difference between factor weight and graded score? It is not clear.

R30: For clarity, we replaced the "graded score" to "the normalized score value of classes for each factor", as shown in P.7, Ln.23-24 of the revised manuscript. The "factor weight" represents the weight of each factor which is determined by the rank of its variance.

C31: Page 8 line 13: why 0.9?

R31: We added a citation (Chen et al., 2005) for the adoption of k=0.9, P.6, Ln.22 in the revised manuscript.

Chen, C. Y., Chen T. C., Yu F. C., Yu W. H., Tseng C. C.: Rainfall duration and debris-flow initiated studies for real-time monitoring, Environ. Geol., 47, 715–724, DOI 10.1007/s00254-004-1203-0, 2005.

C32: Page 8 line 17: Is EAR expressed in mm?

R32: Yes, the unit of EAR is mm.

C33: Page 8 line 21: Is Ir expressed in mm/h?

R33: Yes, the unit of $I_R$ is mm/h.

C34: Page 8 line 26: What is the meaning of rolling hours?

R34: In our study, the average rainfall intensity $I_R$ is calculated in a continuous three hours interval then the calculated time interval moving one hour ahead. For example, the first $I_{3R}$ is the average rainfall intensity for 1-3 hours, then the second $I_{3R}$ is calculated in 2-4 hours, etc.

C35: Page 10 line 6: "thematic map of predisposing factors" instead of "map of the natural environment".

R35: Done, this sentence has been modified in the revised manuscript (P.10, Ln.26).

C36: Page 10 line 9: please make uniform the terminology, as for example causal factors, predisposing factors, impact factors, landslide-inducing factors, etc.

R36: Done, in the revised manuscript, the term "predisposing factors" was used throughout for consistency.

C37: Page 10 lines 21-22-26: please, modify "grid" and "grades".

R37: Done, the terminology has been revised throughout in the revised manuscript.

C38: Page 10 line 27: explicit the values of the six categories.

R38: Based on windward and leeward, the aspects were classified into six categories as following table:

| | |
|---|---|
| South | 157.5° ~ 202.5° |
| Southeast | 112.5° ~ 157.5° |
| Southwest | 202.5° ~ 247.5° |
| East | 67.5° ~ 112.5° |
| West | 247.5° ~ 292.5° |
| Northeast | 22.5° ~ 67.5° |
| Northwest | 202.5° ~ 247.5° |
| North | 337.5° ~ 22.5° |
| Flat | — |

C39: Page 11 line 4: what is the meaning of "geological strength"? The geological map should be classified into classes corresponding to different formations or lithological units.

R39: According to the corresponding compression strengths of the geological lithological properties, this study classified the geological features with such lithological properties by referring to the relationship between compression strength and strength level proposed by ISRM (1981) and conducted level encoding as shown in the following Table.

| Geological term | Characteristics | Strength Level | Class no. |
|---|---|---|---|
| Terrace Accumulation | gravel, clay, soil, sand | extremely weak | 1 |
| Liji layer_Kenting Layer | Foreign rocks in mudstone (badlands terrain) | extremely weak | 1 |
| Alluvial Layer | Soil, sand, gravel | very weak | 2 |
| Takangkou Formation | shale, sandstone, conglomerate rock | weak | 3 |
| Lushan Layer, Sule Layer | Hard shale, slate, phylite, hard sandstone | Medium strong | 4 |
| Bilushan Layer | Shale, quartize sandstone in phylite | Medium strong | 4 |
| Tananao Schists | Black schist , green schist , siliceous schist | strong | 5 |
| Duran Mountain | Agglomerate, tuffaceous sandstone, limestone, convex mirror body | Very strong | 6 |

(modified from ISRM, 1981 and Chen et al., 2009)

ISRM, Rock Characterization Testing and Monitoring-ISRM Suggested Method, Pergamon, London, 1981.

C40: Page 11 line 9: define the analysis function.

R40: Done, more clear information of the analysis function has been added in the revised manuscript (P.11, Ln.30; P.12, Ln.7).

C41: Page 11 lines 10 and 16: explicit the six classes.

R41: The terrain roughness and slope roughness were classified into six categories as shown in the following table:

| Class no. | Terrain roughness | | Class no. | Slope roughness |
|---|---|---|---|---|
| 6 | 367-523 | | 6 | 0-11 |
| 5 | 523-674 | | 5 | 11-20 |
| 4 | 674-950 | | 4 | 20-28 |
| 3 | 950-1035 | | 3 | 28-34 |
| 2 | 1035-1231 | | 2 | 34-41 |
| 1 | 1231-1472 | | 1 | 41-56 |

C42: Page 11 lines 18 and 20: give more information about two factors.

R42: Done, more information was supplemented in the revised manuscript (P.12, Ln.10, Ln.13).

C43: Page 11 line 21: Land disturbance looks like a reclassified land use map. The highest score of disturbance is assigned to bare land, why not to roads and buildings? This is a qualitative attribution, it should be written somewhere.

R43: Based on the tendency to promote landslides, the index of land disturbance was developed (Chen, et al., 2009). The land disturbance in this paper can represent the changes of surface conditions including roads, buildings, crops, bare land, and vegetation. In these factors, we consider bare land has the highest tendency to promote landslide. We supplemented descriptions in the revised manuscript (P.12, Ln.16-17).

**Reviewer 2**

**General comments**

C1: The authors propose in this paper an assessment of landslide susceptibility in a mountain area in Taiwan. The manuscript, which can be interesting for people studying relations between landslide susceptibility and hydrology, has several problems that can be improved after a minor revision. Readers more interested with interactions between natural hazards and roads stay more on the sidelines.

R1: We have confirmed and addressed the reviewer's comments in the revised manuscript. Detailed responses are listed in specific comments and technical corrections, respectively.

**Specific comments**

C2: The state of the art of the methods to evaluate factors influencing landslides in the Introduction is well detailed but can be better structured.

R2: The literature reviews in the Introduction especially landslide susceptibility assessment have been entirely reorganized and rewritten for better readability (P.1, Ln.25 - P.2, Ln.30 in the revised manuscript).

C3: I suggest to add more information / specifications about the study area (surface, length, meters above and under the road path, etc.) and the road (type, traffic, closure consequences, length, history, etc.).

R3: Done, more descriptions of study area and road were supplemented in the revised manuscript (P.9, Ln.12-21).

C4: The presence of the "road" term in the title does not well represent the manuscript content. It should be more focused about the road.

R4: We have modified the title of this paper to "Scale and spatial distribution assessment of rainfall-induced landslides in a catchment with mountain roads" to proper reflect the contents of the paper.

C5: Please define and describe the "landslide" term used in this paper (area, volume, depth, geology, etc.). What kind of landslides do you consider?

R5: In this paper, the term "landslide" is more focus on landslide area. Among the different types of slope failure, debris slides are the easiest and most reliably detected on satellite images in heavily vegetated terrain such as in the mountainous areas of Taiwan. This is because they effectively strip off the vegetation from the slopes, making them readily discernible. Debris slides are, therefore, the major landslides mapped in this study. In cases where vegetation of deep-seated landslides was also stripped off they were also included in the landslide inventory of this study. However, soil slips, soil slides, and debris slides that evolved into debris flows were excluded from our inventory because a clear distinction between the geometry of the source, transport, and depositional areas was not always possible.

C6: There are too much subchapters (2.3.1, 2.3.2, etc.), too much figures and tables in my opinion. I suggest to move some of them in appendices (as Table 9 for instance). Please try to reduce the number of subchapters and keep only the really relevant Figures and Tables for the comprehension of the manuscript.

R6: We removed Table 4 and Table 5, and moved Table 9 to appendices. Besides, the subchapters have been reorganized by using the Arabic numerals 1, 2…etc. instead of sections 2.3.1 and 2.3.2…etc (as shown in P.5, Ln.12, Ln.18; P.6, Ln.6; P.7, Ln.1; P.11, Ln.2; P.13, Ln.1 in the revised manuscript).

**Technical corrections**

C7: Page 1, line 18: mm value for the annually rainfall is wrong, it should be: 2'506 mm and not 2.506 mm.

R7: Done, we corrected this mistake (P.1, Ln.20 in the revised manuscript).

C8: Page 3, line 23: Are really images consistent in quality? Clouds, shadows, etc.

R8: We modified the description (P.3, Ln.4-5 in the revised manuscript).

C9: Page 3, line 23: Studies have indicated… Which studies?

R9: We added the references (P.3, Ln.7 in the revised manuscript).

C10: Page 3, line 1: The results of this study could serve as a reference… Maybe too presumptuous.

R10: We removed this sentence in the revised manuscript (P.3, Ln.18-19).

C11: Page 4, line 18: Please define x.

R11: Actually, a few of variable "X" was mistyped as "x". We corrected this mistake in the revised manuscript (P.4, Ln.4, Ln.13).

C12: Page 5, line 5: Please define k.

R12: We have added the definition of $k$ in the revised manuscript. (P.4, Ln.21).

C13: Page 6, line 9: Please define d.

R13: We have added the definition of $d$ in the revised manuscript. (P.7, Ln.23-24 in the revised manuscript).

C14: Page 6, line 25: In the table: Please add the Table number.

R14: Done, we added the table number (P.5, Ln.5 in the revised manuscript).

C15: Page 7, line 5: Please give the complete name of OA.

R15: Done, we added the complete name of OA (P.5, Ln.12 in the revised manuscript).

C16: Page 7, line 15: Please give explanation of $X_{i+}$, $X_{+i}$, $X_{ii}$.

R16: Done, we added the explanation of $X_{i+}$, $X_{+i}$ and $X_{ii}$, respectively (P.5, Ln.16-17, Ln.24-25 in the revised manuscript).

C17: Page 7, line 23: Please give the complete name of EAR.

R17: Done, we added the complete name of EAR (P.6, Ln.6 in the revised manuscript).

C18: Page 8, line 20: Please give the complete name of $I_R$.

R18: Done, we added the complete name of $I_R$ (P.7, Ln.1 in the revised manuscript).

C19: Page 8, line 25 and 26: contradiction between I (rainfall intensity) and $I_R$ (not explained).

R19: In our study, "I" denotes hourly rainfall (60 minutes intensity), intensity of rolling rainfall "$I_R$" represent summation of selected 60 minutes intensity (I) during m rolling hours.

C20: Page 9, line 14-15: Please reword the sentence.

R20: Done, we simplified this sentence in the revised manuscript (P.10, Ln.5-6).

C21: Page 9, 4.1, please give image info's (resolution, surface, etc.).

R21: Done, we supplemented FORMOSAT-2 image description in the revised manuscript (P.9, Ln.26 – P.10, Ln.2).

C22: Page 9, line 25: different interpretation factors: which ones?

R22: we selected areas with water, roads, buildings, crops, vegetation, river channels, and bare land within the study area as the sample area factors for interpretation training.

C23: Page 10, line 4: why 8 x 8 m (and not 10 x 10 m or 5 x 5 m)?

R23: Since the raw spatial resolution of FORMOSAT-2 (FM2) images is 8 x 8 m, we prepared 8 x 8 m thematic map of predisposing factors, as well as 8 x 8 m DEM to construct landslide susceptibility map.

C24: -Page 10, line 5: we also constructed an … DEM: how?

R24: We downgraded from the 5 x 5 m MOI (Ministry of the Interior, Taiwan) DEM to obtain the 8 x 8 m DEM used in this study.

C25: Page 10, line 16: 1480.6 and 365.2 m: are the values after the dot really needed?

R25: We modified these two number to 1481 and 365, respectively. It would not affect the classification results (P.11, Ln.9 in the revised manuscript).

C26: Page 10, line 22: seven grades: why seven, for what reason?

R26: According to the classifications of gradient for hillslope land use limit proposed by SWCB (2017), the gradient can be classified into six grades as following left table. We modified gradient interval of grade 6 as "55-100", and added additional grade with gradient interval of ">100" as shown in the following right table used in our study.

| Classifications of gradient (SWCB) | Gradient (%) |
|---|---|
| 1st | <5 |
| 2nd | 5-15 |
| 3rd | 15-30 |
| 4th | 30-40 |
| 5th | 40-55 |
| 6th | >55 |

| Class no. | Gradient (%) |
|---|---|
| 7 | <5 |
| 6 | 5-15 |
| 5 | 15-30 |
| 4 | 30-40 |
| 3 | 40-55 |
| 2 | 55-100 |
| 1 | >100 |

Soil and Water Conservation Bureau (SWCB), Council of Agriculture (COA), Executive Yuan, R.O.C. (Taiwan), https://www.swcb.gov.tw/class2/index.asp?ct=laws&m1=10&m2=55&AutoID=22, 2017 (in Chinese).

C27: Page 10, line 23: seven grades: why six, for what reason?

R27: Based on windward and leeward, the aspects were classified into six categories as following table:

| | |
|---|---|
| South | 157.5° ~ 202.5° |
| Southeast | 112.5° ~ 157.5° |
| Southwest | 202.5° ~ 247.5° |
| East | 67.5° ~ 112.5° |
| West | 247.5° ~ 292.5° |
| Northeast | 22.5° ~ 67.5° |
| Northwest | 202.5° ~ 247.5° |
| North | 337.5° ~ 22.5° |
| Flat | — |

C28: Page 10, line 27: six categories: why six, for what reason?

R28: Same as C27 and R27.

C29: Page 11, line 5: six grades: why six, for what reason?

R29: According to the corresponding compression strengths of the geological lithological properties, this study classified the geological features with such lithological properties by referring to the relationship between compression strength and strength level proposed by ISRM (1981) and conducted level encoding as shown in the following Table.

| Geological term | Characteristics | Strength Level | Class no. |
|---|---|---|---|
| Terrace Accumulation | gravel, clay, soil, sand | extremely weak | 1 |
| Liji layer_Kenting Layer | Foreign rocks in mudstone (badlands terrain) | extremely weak | 1 |
| Alluvial Layer | Soil, sand, gravel | very weak | 2 |
| Takangkou Formation | shale, sandstone, conglomerate rock | weak | 3 |
| Lushan Layer, Sule Layer | Hard shale, slate, phylite, hard sandstone | Medium strong | 4 |
| Bilushan Layer | Shale, quartize sandstone in phylite | Medium strong | 4 |
| Tananao Schists | Black schist , green schist , siliceous schist | strong | 5 |
| Duran Mountain | Agglomerate, tuffaceous sandstone, limestone, convex mirror body | Very strong | 6 |

(modified from ISRM, 1981 and Chen et al., 2009)

ISRM, Rock Characterization Testing and Monitoring-ISRM Suggested Method, Pergamon, London, 1981.

C30: Page 11, line 10: six grades: why six, for what reason?

R30: The terrain roughness was classified into six categories by using cluster analysis of SPSS software as shown in the following table:

| Class no. | Terrain roughness |
|---|---|
| 6 | 367-523 |
| 5 | 523-674 |
| 4 | 674-950 |
| 3 | 950-1035 |
| 2 | 1035-1231 |
| 1 | 1231-1472 |

C31: Page 11, line 16: six grades: why six, for what reason?

R31: The slope roughness was also classified into six categories by using cluster analysis of SPSS software as shown in the following table:

| Class no. | Slope roughness |
|---|---|
| 6 | 0-11 |
| 5 | 11-20 |
| 4 | 20-28 |
| 3 | 28-34 |
| 2 | 34-41 |
| 1 | 41-56 |

C32: Page 11, line 18: seven grades: why seven, for what reason?

R32: The distance to water was classified into seven categories as shown in the following table:

| Class no. | Distance to water (m) |
|---|---|
| 1 | <100 |
| 2 | 100-300 |
| 3 | 300-500 |
| 4 | 500-700 |
| 5 | 700-1000 |
| 6 | 1000-1500 |
| 7 | >1500 |

C33: Page 11, line 20: seven grades: why seven, for what reason?

R33: The distance to road was classified into seven categories as shown in the following table:

| Class no. | Distance to road (m) |
|---|---|
| 1 | <100 |
| 2 | 100-300 |
| 3 | 300-500 |
| 4 | 500-700 |
| 5 | 700-1000 |
| 6 | 1000-1500 |
| 7 | >1500 |

C34: Page 12, line 2: six grades: why six, for what reason?

R34: The rainfalls were also classified into six categories by using cluster analysis of SPSS software as shown in the following table:

| Class no. | EAR (mm) | Max $I_{3R}$ (mm/h) |
|---|---|---|
| 6 | 285-302 | 59-61 |
| 5 | 302-313 | 61-62 |
| 4 | 313-321 | 62-64 |
| 3 | 321-329 | 64-65 |
| 2 | 329-340 | 65-67 |
| 1 | 340-363 | 67-68 |

C35: Page 12, line 7: SPSS : maybe add "software" to more better describe what it is.

R35: Done, we added "software" following SPSS (P.13, Ln.5 in the revised manuscript).

C36: Page 12, line 25: seven grades: why seven, for what reason?

R36: The elevation data was divided into seven grades at intervals of 300 m, as shown in the following table:

| Class no. | Elevation (m) |
|---|---|
| 7 | <450 |
| 6 | 450-750 |
| 5 | 750-1050 |
| 4 | 1050-1350 |
| 3 | 1350-1650 |
| 2 | 1650-1950 |
| 1 | >1950 |

C37: Page 13, lines 3 and 4: please clarify the sentence with the values in " () " : 2.02 and 9.96 = $I_{3R}$.

R37: For the clarity, we modified the descriptions in the revised manuscript (P.14, Ln.2-3).

C38: Page 13, line 20: four level: why this repartition and not 0-25, 25-50, 50-75 and 75-100?

R38: We divided landslide susceptibility into four levels: high (0.731–1), medium high (0.461–0.73), medium low (0.23–0.46), and low (0–0.23) are based on the mean probability of landslide occurrence to be 0.46.

C39: All figure and table captions: please verify that every caption is ended by a ".".

R39: Done, all captions were checked and corrected throughout in the revised manuscript.

C40: Page 24, caption Figure 2: which "blue line" do you mean? Please try to redo the image (for example the "t" Toayan district, is not well readable), colours are no well appropriated.

R40: Done, Figure 2 has been redrawn to strengthen visibility (P.27 in the revised manuscript). In additions, the distribution of mountain roads is represented by purple lines, the mistake was corrected.

C41: Page 35, Table 2: where are the "before" and "after" data in the error matrix (lines or columns)? Please clarify.

R41: Table 2 represents error matrix of interpretation results of satellite images "after" Typhoon Kong-rey, we corrected this mistake (P.38 in the revised manuscript)

C42: Page 38: Table 5: table not necessary / relevant for the paper

R42: Done, we removed Table 4 and Table 5 in the revised manuscript.

C43: Page 39: Table 6: please give units.

R43: Done, we added units of EAR and $I_{3R}$ in Table 6 (P.42 in the revised manuscript).

C44: Page 44: Table 10: please define $D_{t,min}$ and $D_{t,max}$.

R44: The $D_{t,min}$ and $D_{t,max}$ represent minimum and maximum value of instability index (Eq. (8) in the original version; Eq. (13) in the revised version), respectively.

[revised manuscript text omitted]

---

## Author Response (AR2)

Dear Editor,

We deeply appreciate your efforts in evaluating our manuscript. We provided reference of DEM and added descriptions of landslide inventory, as marked yellow in P.10 in the revised manuscript.

With the best regards,

Chih-Ming Tseng, Yie-Ruey Chen, Szu-Mi Wu